# Endothelial deletion of SHP2 suppresses tumor angiogenesis and promotes vascular normalization

Zhiyong Xu [1,2], Chunyi Guo[1], Qiaoli Ye[1], Yueli Shi[2], Yihui Sun[1], Jie Zhang[3], Jiaqi Huang [1], Yizhou Huang[4], Chunlai Zeng[5], Xue Zhang [1,6], Yuehai Ke [1,6,7 ✉] & Hongqiang Cheng [1,8 ✉]

SHP2 mediates the activities of multiple receptor tyrosine kinase signaling and its function in endothelial processes has been explored extensively. However, genetic studies on the role of SHP2 in tumor angiogenesis have not been conducted. Here, we show that SHP2 is activated in tumor endothelia. *Shp2* deletion and pharmacological inhibition reduce tumor growth and microvascular density in multiple mouse tumor models. *Shp2* deletion also leads to tumor vascular normalization, indicated by increased pericyte coverage and vessel perfusion. SHP2 inefficiency impairs endothelial cell proliferation, migration, and tubulogenesis through downregulating the expression of proangiogenic SRY-Box transcription factor 7 (SOX7), whose re-expression restores endothelial function in SHP2-knockdown cells and tumor growth, angiogenesis, and vascular abnormalization in *Shp2*-deleted mice. SHP2 stabilizes apoptosis signal-regulating kinase 1 (ASK1), which regulates SOX7 expression mediated by c-Jun. Our studies suggest SHP2 in tumor associated endothelial cells is a promising anti-angiogenic target for cancer therapy.

[1] Department of Pathology and Pathophysiology and Sir Run Run Shaw Hospital, Zhejiang University School of Medicine, Hangzhou, China. [2] The Fourth Affiliated Hospital, Zhejiang University School of Medicine, Yiwu, China. [3] Department of Urology of Sir Run Run Shaw Hospital, Zhejiang University School of Medicine, Hangzhou, China. [4] Department of Gynecology of Women's Hospital, Zhejiang University School of Medicine, Hangzhou, China. [5] Department of Cardiology, Lishui Hospital of Zhejiang University, The Fifth Affiliated Hospital of Wenzhou Medical University, Lishui Municipal Central Hospital, Lishui, China. [6] Department of Respiratory Medicine of Sir Run Run Shaw Hospital, Zhejiang University School of Medicine, Hangzhou, China. [7] Cancer Center, Zhejiang University, Hangzhou, China. [8] Department of Cardiology of Sir Run Run Shaw Hospital, Zhejiang University School of Medicine, Hangzhou, China. ✉email: yke@zju.edu.cn; hqcheng11@zju.edu.cn

Tumor growth, similar to that of normal organs, depends greatly on the outgrowth of blood vessels to form neo-vasculatures that supply raw materials for metabolism. Neovascular structures originate from existing capillaries via angiogenesis[1], which is a robust process that involves the establishment of a dynamic balance of pro- and anti-angiogenic factors suspended in the tissue niche. VEGF-A is a critical proangiogenic factor that renders the development of anti-VEGF therapies a promising avenue of research, particularly in the treatment of cancers[2]. Inhibitors targeting the VEGF pathway have exhibited significant anti-tumor activities in multiple mouse models; however, anti-VEGF therapies have not yet been successfully clinically applied to most cancers in humans[3,4]. In the absence of VEGF, other proangiogenic factors, including FGF, and ephrin, regulate angiogenesis in anti-VEGF-treated tumors and it has been found that anti-VEGF therapy induces unexpected vessel normalization and extracellular matrix remodeling, with improved vessel perfusion[5,6]. Vascular abnormalities in cancers include insufficient pericyte coverage, hyperpermeability, and immunosurveillance escape[7,8]. The challenges faced by novel anti-angiogenesis therapy in cancers have led to rigorous investigations, particularly in the elucidation of the underlying mechanisms.

The protein tyrosine phosphatase SHP2, encoded by *PTPN11*, is expressed ubiquitously, including in blood vessel cells. As an important participant in growth factor and cytokine signaling, SHP2 is frequently upregulated or mutated in tumors, and its oncogenic behavior has been attributed to abnormal SHP2 expression[9–11]. Structurally, there are two SH2 domains, a protein tyrosine phosphatase (PTP) domain, and a C-terminal tail containing phosphorylatable tyrosine residues. The SH2 domains interact with the PTP domain to form a self-inhibitory intramolecular interaction, and have recently been used to develop allosteric inhibitors[12,13]. New SHP2 inhibitors have been demonstrated to possess tremendous anti-tumor activity, particularly in combination with other conventional drugs[14–20]. Owing to the ubiquitous expression of druggable SHP2, its function in tumor microenvironments has been studied thoroughly. Our group has demonstrated that SHP2 suppressed CXCL9 production in macrophages, which prevented T cell infiltration and promoted tumor growth in mice[21]. In addition, SHP2 mediates PD-1 signaling in tumor-infiltrating lymphocytes and is important for tumor immunosuppression[22–26]. These findings suggest that SHP2 plays important tumor-intrinsic and -extrinsic functions.

SHP2 has been reported to interact directly with VEGFR2; thus, it is an important regulator of VEGF-VEGFR2 signaling[27,28]. In contrast, multiple receptors, including the dopamine, collagen I, AXL, and Mer receptors, recruit and activate SHP2 when activated by their ligands to inhibit VEGFR signaling and endothelial function[29–31]. These studies demonstrated that SHP2 functioning in endothelial cells is complicated. SHP2 interacts with VE-cadherin and regulates its phosphorylation, which is associated with VEGFR signaling and affects the permeability of the endothelial barrier[32,33]. Our previous study using endothelium-selective Shp2 knockout mice indicated that SHP2 plays an essential role in vascular development and homeostasis maintenance via the regulation of endocytosis-mediated VE-cadherin cycling[34]. GRB2-associated binding protein 1 (GAB1), a scaffolding protein for SHP2, has been found to promote postnatal angiogenesis[35–37]. Another scaffolding protein, annexin A2, facilitates SHP2 in the dephosphorylation of VE-cadherin in angiogenesis[38,39]. The anti-angiogenic activity of SHP2 inhibitors has recently been suggested in mouse tumor models[40,41]; however, the direct genetic evidence to demonstrate the role of SHP2 in angiogenesis, particularly in tumor angiogenesis, require investigation.

Using an endothelial-cell-specific and inducible knockout strategy to delete SHP2 and multiple mouse tumor models, we find that SHP2 is a key regulator in tumor angiogenesis and vessel abnormalization. As SHP2 is central to multiple proangiogenic signaling pathways and its allosteric inhibitors are relatively easy to access, SHP2 is a promising target for anti-angiogenic therapies in cancers.

## Results

**SHP2 is activated in tumor endothelial cells.** The phosphorylation of the tyrosine residues (Y542) at the SHP2 C-terminal tail is an indicator of SHP2 activation. Conditioned media from different non-small cell lung cancer cells (NSCLC) induced a dramatic increase in levels of both phosphorylated and total SHP2 in human umbilical vein endothelial cells (HUVECs; Fig. 1a). Next, clinical lung cancer and adjacent normal lung tissues were collected for immunofluorescence staining. The level of p-SHP2 in platelet endothelial cell adhesion molecule-1 (CD31)-labeled tumor endothelial cells was markedly higher than that in control vessels (Fig. 1b). Similarly, SHP2 expression was slightly, but significantly, increased in tumor vessels (Fig. 1c). Finally, the GSE118904 dataset[42], generated by utilizing mouse tumor endothelial and normal endothelial cells for single-cell RNA sequencing, was used for analysis, and the level of SHP2 was increased in tumor endothelial cells (Fig. 1d). These results indicate that SHP2 was increased and significantly activated in tumor endothelial cells.

**Shp2 deletion in endothelial cells inhibits tumor growth.** To define the function of SHP2 in tumor endothelial cells, endothelial-cell-specific and inducible Shp2 knockout mice (Shp2iECKO) were introduced. *Shp2* was deleted effectively after tamoxifen induction (Supplementary Fig. 1a). One week after the final tamoxifen injection, syngeneic Lewis lung cancer (LLC) cells were implanted subcutaneously into the flank of Shp2iECKO and control mice. Compared with the control mice, tumor growth was remarkably suppressed in Shp2iECKO mice (Fig. 1e–g). We observed similar inhibition in tumor growth in Shp2iECKO mice in a second syngeneic tumor model using B16 melanoma cells (Supplementary Fig. 1b–d). In addition, *Shp2* deletion also reduced tumor growth in an orthotopic tumor model utilizing E0771 breast cancer cells (Fig. 1h–j). Moreover, *Shp2* deletion led to a significant reduction in the necrotic, apoptotic, and hypoxic areas in tumors (Fig. 1k–m). Our data based on multiple mouse tumor models strongly suggest that endothelial SHP2 is essential for tumor growth.

**Shp2 deletion impairs tumor angiogenesis.** To elucidate the mechanisms by which endothelial *Shp2* deletion inhibited tumor growth, the tumor microvasculature in Shp2iECKO and control mice was examined. CD31-labeled microvessels were decreased in Shp2iECKO mice for all three tumor models (Fig. 2a and Supplementary Fig. 1e, f). LLC cells expressed more proangiogenic factors, such as VEGF-A, FGF1, and FGF2, than B16 cells (Supplementary Fig. 1g, h), resulting in the formation of a highly dense vasculature. To exclude the possibility that our observations were due to the interaction between tumor endothelial and cancer cells, a Matrigel plug assay was conducted (Fig. 2b). The plugs in the control mice exhibited red coloration, indicating new vessel formation; while those in Shp2iECKO mice were pale, indicating that fewer new vessels were formed (Fig. 2b). Concordantly, there were significantly fewer CD31-positive cells in the plugs obtained from SHP2iECKO mice than those obtained from control mice (Fig. 2c); moreover, the deletion of *Shp2* in endothelial cells markedly reduced the formation of microvessels from the aortic rings (Fig. 2d). Thus, SHP2 in endothelial cells is necessary for in vivo angiogenesis.

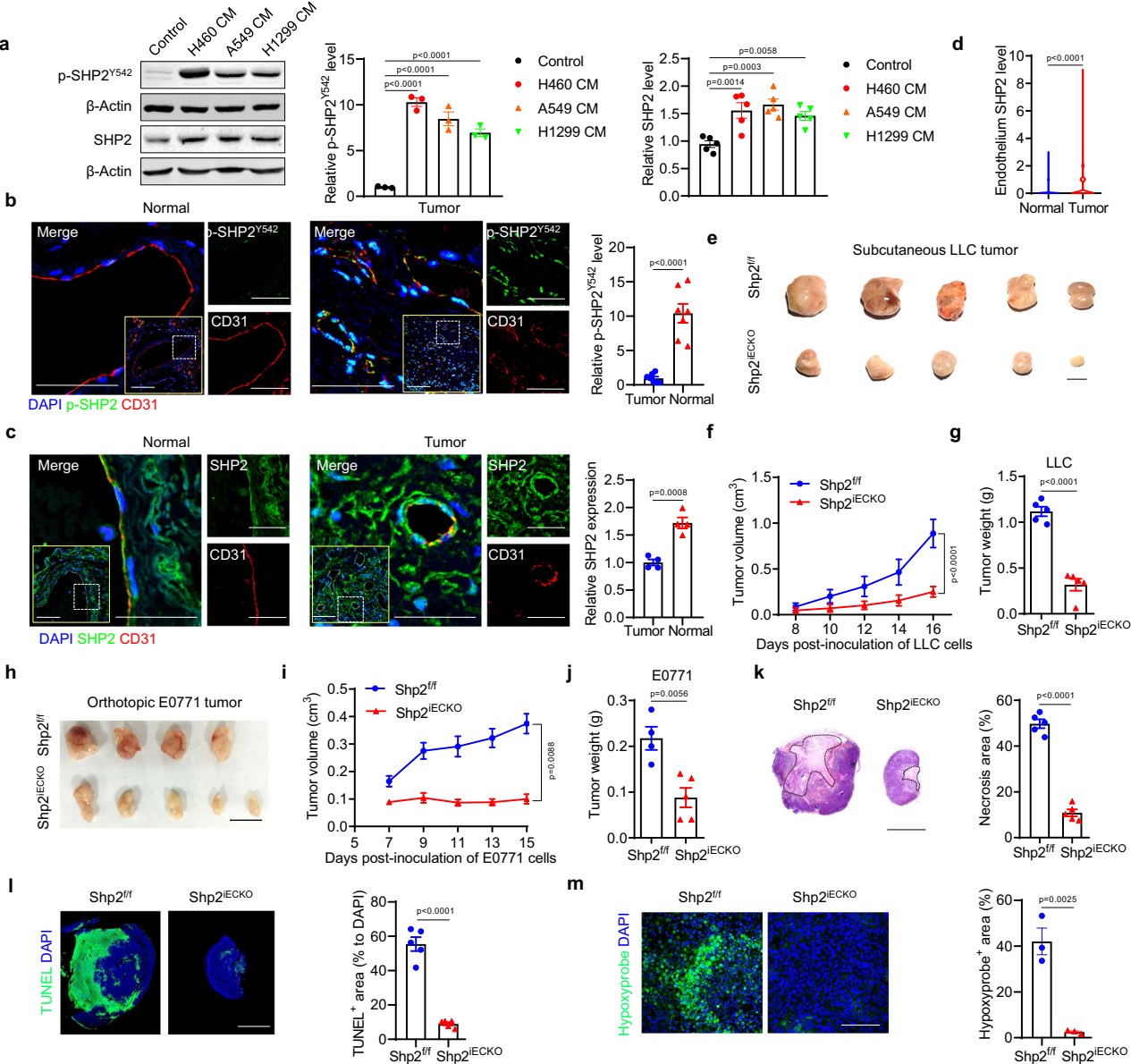

**Fig. 1 Hyper-activated SHP2 in tumor endothelial cells is necessary for tumor growth. a** Western blot for p-SHP2$^{Y542}$ and SHP2 in HUVECs treated with various NSCLC cancer cell-conditioned media for 10 min (p-SHP2, $n = 3$) or 24 h (SHP2, $n = 5$). β-Actin was used as a loading control. Quantitative data were shown as mean ± SEM. $p$-values were shown and generated by one-way ANOVA with multi-comparisons. **b, c** Representative images showing p-SHP2$^{Y542}$ (**b**, green) and SHP2 expression (**c**, green) in CD31$^+$ vessels (red) in NSCLC tumor tissues and paired adjacent normal tissues ($n = 7$ for p-SHP2 and $n = 4$ for SHP2). Quantitative data were shown as mean ± SEM. $p$-values were shown and generated by using the two-tailed Student's $t$-test. Scale bar: 50 μm; inset 100 μm. **d** *Shp2* mRNA expression in mouse normal and tumor endothelial cells was extracted from the GEO dataset (GSE118904, $n = 1000$ cells). $p$-value was shown and generated by using the two-tailed Student's $t$-test. **e** Images of explant LLC tumors from Shp2$^{f/f}$ and Shp2$^{iECKO}$ mice. Scale bar: 10 mm. **f, g** The volumes and weights of LLC tumors in Shp2$^{f/f}$ ($n = 5$) and Shp2$^{iECKO}$ ($n = 5$) mice. The weights were recorded 16 days after cancer cell injection. Quantitative data were shown as mean ± SEM. $p$-values were shown and generated by two-way ANOVA with Tukey's post hoc test (**f**) or by using the two-tailed Student's $t$-test (**g**). **h** Images for orthotopic E0771 tumors in Shp2$^{f/f}$ ($n = 4$) and Shp2$^{iECKO}$ ($n = 5$) mice. Scale bar: 10 mm. **i, j** The volumes (**i**) and weights (**j**) for orthotopic E0771 tumors in Shp2$^{f/f}$ ($n = 4$) and Shp2$^{iECKO}$ ($n = 5$) mice. Tumor weights were measured 15 days after cancer cell injection. Quantitative data were shown as mean ± SEM. $p$-values were shown and generated by two-way ANOVA with Tukey's post hoc test or by using the two-tailed Student's $t$-test. **k, l** Representative images of LLC tumor necrosis (**k**) and apoptosis (**l**). H&E staining was conducted and necrosis area was labeled by dot lines. Scale bar: 5 mm. Apoptosis was measured by TUNEL staining. Scale bar: 5 mm. DAPI was used to label nuclei. Quantitative data were shown as mean ± SEM. $n = 5$ for each group. $p$-values were shown and generated by using the two-tailed Student's $t$-test. **m** Representative images of hypoxyprobe-1-labeled areas in LLC tumors. Quantitative data were shown as mean ± SEM. $n = 3$ for each group. $p$-value was shown and generated by using the two-tailed Student's $t$-test. Scale bar: 100 μm. Source data are provided as a Source data file.

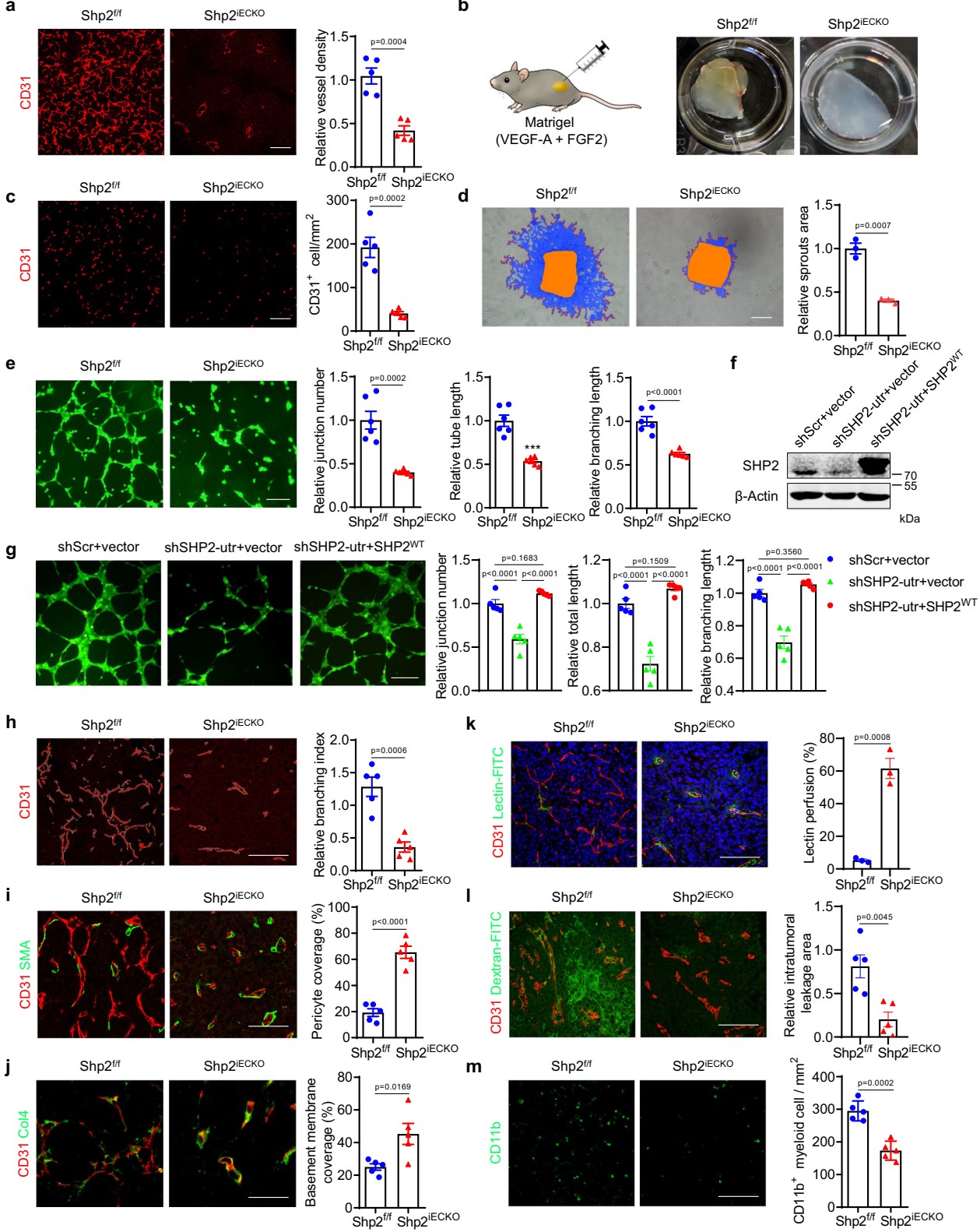

To further explore the role of SHP2 in angiogenesis, lung endothelial cells were isolated from Shp2[iECKO] and control mice. Meanwhile, HUVECs and hCMECs were transduced with short hairpin RNA (shRNA) to knockdown SHP2 or treated with an SHP2 inhibitor, RMC4550. *Shp2* deletion impaired endothelial cell proliferation and migration, as indicated by the EdU incorporation and transwell cell migration assays, as well as tube formation, characterized by the decrease in the number of junctions, tube lengths, and branching lengths (Fig. 2e and Supplementary Fig. 2a, b). We observed the same defects in cell proliferation, migration, and tube formation in HUVECs upon SHP2 inhibition with various RMC4550 concentrations, and in SHP2-knockdown HUVECs and hCMECs (human cerebral microvessel endothelial cells) (Fig. 2f, g and Supplementary

**Fig. 2 Shp2 deletion in endothelial cells impairs tumor angiogenesis and promotes vascular normalization. a** Immunofluorescence images of CD31 in LLC tumors in Shp2$^{f/f}$ ($n = 5$) and Shp2$^{iECKO}$ ($n = 5$) mice. Quantitative data were shown as mean ± SEM. *p*-value was shown and generated by using the two-tailed Student's *t*-test. Scale bar: 100 μm. **b** Images of Matrigel plugs from Shp2$^{f/f}$ and Shp2$^{iECKO}$ mice. **c** Immunofluorescence images of CD31 in plugs from Shp2$^{f/f}$ ($n = 5$) and Shp2$^{iECKO}$ ($n = 5$) mice. Images were analyzed by the Image J software and quantitative data were shown as mean ± SEM. *p*-value was shown and generated by using the two-tailed Student's *t*-test. Scale bar: 100 μm. **d** Sprouting assay of aortic rings from Shp2$^{f/f}$ ($n = 3$) and Shp2$^{iECKO}$ ($n = 3$) mice. Quantitative sprouting area were shown as mean ± SEM. *p*-value was shown and generated by using the two-tailed Student's *t*-test. Scale bar: 100 μm. **e** Tube formation in vitro of MLECs isolated from Shp2$^{f/f}$ ($n = 5$) and Shp2$^{iECKO}$ ($n = 5$) mice. Junction numbers, tube lengths, and branching lengths were measured by the Image J software and were shown as mean ± SEM. *p*-values were shown and generated by using the two-tailed Student's *t*-test. Scale bar: 100 μm. **f** Western blot for SHP2 expression in SHP2-knockdown HUVECs with SHP2$^{WT}$ re-expression. β-Actin was used as a loading control. Results were repeated for three independent experiments. **g** Tube formation in vitro in SHP2-knockdown HUVECs with SHP2 re-expression. Junction numbers, tube lengths, and branching lengths were measured by using the Image J software and shown as mean ± SEM. *p*-values were shown and generated by using one-way ANOVA with multi-comparisons. $n = 5$ for each group. Scale bar: 100 μm. **h–j** Immunofluorescence images of CD31 (**h**), αSMA (**i**), Collagen IV (**j**, Col4) in LLC tumors from Shp2$^{f/f}$ ($n = 5$) and Shp2$^{iECKO}$ ($n = 5$) mice. Branching index, pericyte coverage, basement membrane coverage were measured by using Image J software and quantitative data were shown as mean ± SEM. *p*-values were shown and generated by using the two-tailed Student's *t*-test. Scale bar: 100 μm. **k**, **l** Representative images for vessels perfused with lectin (**k**) and dextran (**l**) in LLC tumors from Shp2$^{f/f}$ ($n = 5$) and Shp2$^{iECKO}$ ($n = 5$) mice. Images were analyzed by the Image J software and Quantitative data were shown as mean ± SEM. *p*-values were shown and generated by using the two-tailed Student's *t*-test. Scale bar: 100 μm. **m** Immunofluorescence images of CD11b$^+$ myeloid cells in LLC tumors from Shp2$^{f/f}$ ($n = 5$) and Shp2$^{iECKO}$ ($n = 5$) mice. Images were analyzed by the Image J software and Quantitative data were shown as mean ± SEM. *p*-values was shown and generated by using the two-tailed Student's *t*-test. Scale bar: 100 μm. Source data are provided as a Source data file.

Fig. 2c-k). Notably, SHP2 re-expression in SHP2-knockdown cells restored endothelial cell proliferation, migration, and tube formation (Fig. 2f, g and Supplementary Fig. 2f-k). Taken together, SHP2 is essential for angiogenesis in vitro.

In addition, *Shp2* deletion in the adult mouse endothelium did not affect the histological morphologies and microvascular density of various organs, including the lungs, spleens, kidneys, livers, and hearts (Supplementary Fig. 3a, b). Furthermore, *Shp2* deletion in adult mouse endothelia did not change the microvascular permeability in the lungs, kidneys, and livers of mice, even with tumor burden (Supplementary Fig. 3c, d). Endothelial *Shp2* deletion in adult mice did not disturb the basal vascular homeostasis.

**Shp2 deletion results in tumor vascular normalization.** We further characterized LLC tumor vessels in Shp2$^{iECKO}$ mice, which were less branched than those in control mice (Fig. 2h), and their α-smooth muscle actin (αSMA) positive pericyte coverage along tumor vessels was higher (Fig. 2i). Moreover, the level of collagen IV, an integral component of the endothelial cell basement membrane (BM), was also increased in the tumor vessels of Shp2$^{iECKO}$ mice (Fig. 2j). Defects in vascular branching, pericyte coverage, and BM coverage were also observed in the B16 and E0771 tumors in Shp2$^{iECKO}$ mice (Supplementary Fig. 4a-f). PDGF-BB, an important regulator of vascular maturation, was upregulated in SHP2-knockdown endothelial cells, which might explain the increase in pericyte and BM coverage in the tumor vessels of Shp2$^{iECKO}$ mice (Supplementary Fig. 4g). Moreover, the vascular abnormalization featuring hyperpermeability and poor vessel perfusion in the tumor vessels of control mice was markedly reduced in Shp2$^{iECKO}$ mice (Fig. 2k, l). The improved perfusion in the tumor vessels of Shp2$^{iECKO}$ mice decreased the infiltration of CD11b positive myeloid cells (Fig. 2m). Thus, *Shp2* deletion in tumor endothelial cells promoted tumor vascular normalization.

**SHP2 inhibitor SHP099 impairs tumor growth and angiogenesis.** *Shp2* deletion in endothelial cells greatly impaired tumor growth and angiogenesis, while promoting vascular normalization. Many allosteric SHP2 inhibitors, including SHP099, a prototype, have been developed and have played promising anti-tumor roles in preclinical experiments and clinical trials. To characterize the anti-angiogenic effect of SHP099, inhibitor-resistant SHP2 (SHP2$^{T253M/Q257L}$) was constructed and expressed in SHP2-knockdown LM3 hepatic cancer cells. As expected, SHP099

(75 mg/kg) significantly inhibited tumor growth in the LM3 tumors of nude mice (Fig. 3a–d). Similarly, SHP099 (75 mg/kg) inhibited the growth of LM3 tumors that expressed the SHP099-resistant SHP2 mutant (Fig. 3a–d). Our results are consistent with those of two recent studies demonstrating the anti-angiogenic role of SHP2 inhibitors[40]. Tumor vessels were then characterized, and SHP099 decreased the microvessel density, vascular branching, and BM coverage of tumors (Fig. 3e, f, h, i, j, l). In contrast to that with *Shp2* deletion, SHP099 significantly decreased the coverage of pericyte (Fig. 3g, k), indicating that SHP2 in pericytes is required for recruitment. In addition, Cediranib, a VEGFR inhibitor, also inhibited LM3 tumor growth (Fig. 3a–d), indicating that endothelial cell-dependent angiogenesis is required for LM3 tumor growth. Our results are consistent with recently published results and strongly suggested that endothelial-SHP2-mediated angiogenesis greatly contributes to tumor growth[40,41].

**SHP2 regulates the expression of proangiogenic SOX7.** To determine the molecular mechanisms underlying SHP2-mediated tumor angiogenesis, angiogenesis-related genes were analyzed in SHP2-knockdown and control HUVECs. The expression of most genes was unaltered or increased after SHP2-knockdown, while the expression of two genes, i.e., SOX7 and SOX18, decreased significantly (Fig. 4a). SHP2-knockdown and inhibition inhibited angiogenesis; therefore we focused on the downregulated genes. Western blotting consistently showed that SOX7 and SOX18 were reduced upon SHP2 knockdown in HUVECs (Fig. 4b and Supplementary Fig. 5a, b). SOX17 expression was unchanged with SHP2 knockdown (Supplementary Fig. 5b). SOX7, SOX17, and SOX18 belong to the SOXF family and are functionally redundant; moreover, SOX7 deletion in endothelial cells reduces tumor angiogenesis and promotes tumor vascular normalization[43]. Thus, we selected SOX7 for further studies. Its expression was also decreased in either lung endothelial cells from Shp2$^{iECKO}$ mice or HUVECs treated with the SHP2 inhibitor RMC4550 (Supplementary Fig. 5c, d). Immunofluorescence staining indicated a similar decrease in SOX7 expression in SHP2-knockdown or RMC4550-treated HUVECs (Supplementary Fig. 5e). In contrast, the overexpression of wild-type or an active mutant SHP2 (A72G) enhanced SOX7 expression (Fig. 4c). Similar to SHP2, conditioned media obtained from the cancer cell culture induced SOX7 expression (Fig. 4d). In HUVECs, VEGF-A or FGF2 induced SOX7 expression, but not that of SOX18, in a time-dependent manner, which was remarkably suppressed by SHP2-knockdown or SHP2

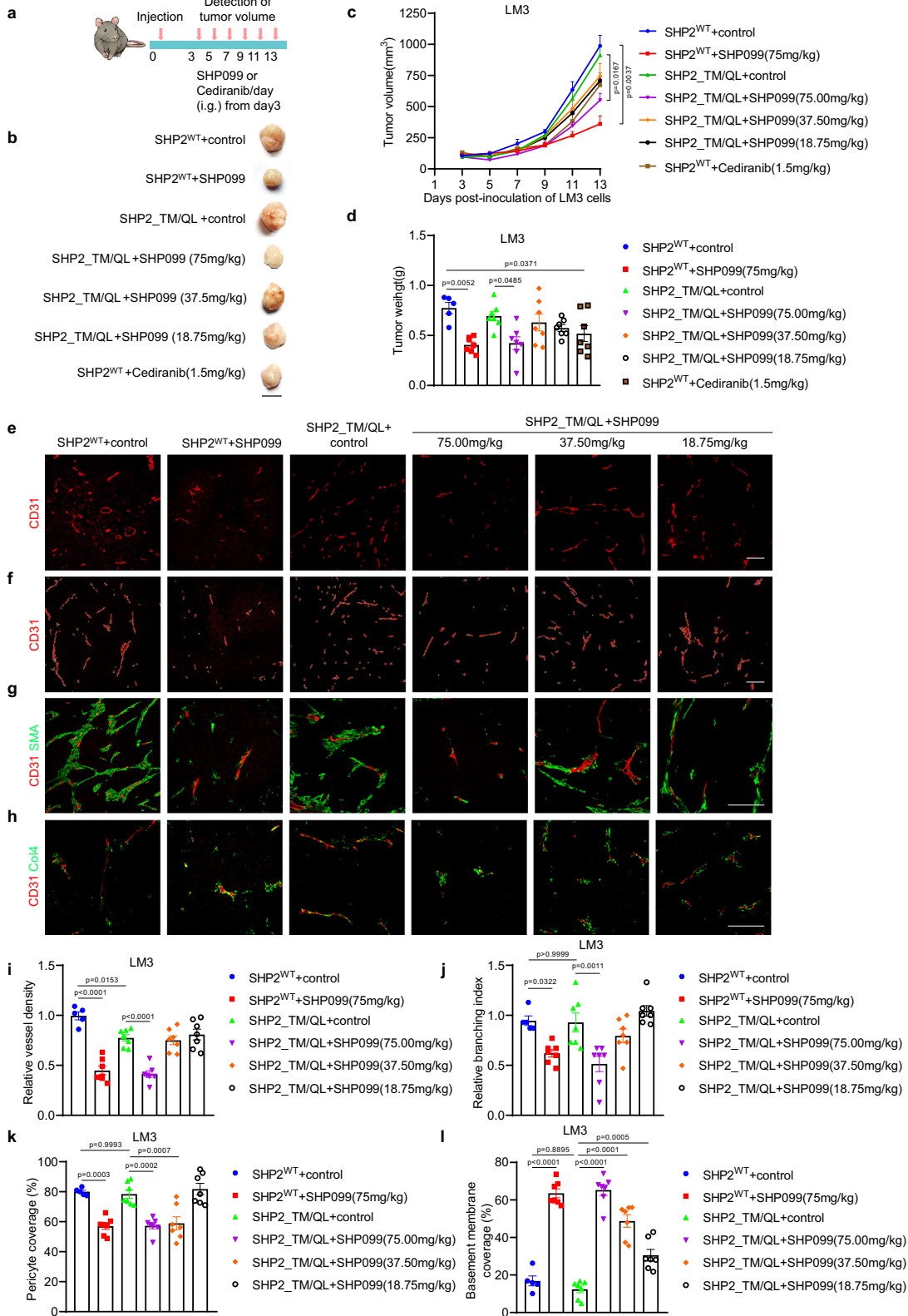

6    NATURE COMMUNICATIONS | (2021)12:6310 | https://doi.org/10.1038/s41467-021-26697-8 | www.nature.com/naturecommunications

inhibition (Fig. 4e–h and Supplementary Fig. 5f-h). In addition, the immunofluorescence staining results showed that SOX7 expression in the LLC tumors and plugs of Shp2$^{iECKO}$ mice was lower than that in control mice (Fig. 4i, and Supplementary Fig. 5i). Similar to SHP2, SOX7 was increased in CD31-positive endothelial cells in NSCLC tissues than that in normal tissues (Fig. 4j). Therefore, SOX7 expression in endothelial cells is controlled by SHP2.

**SOX7 is required for SHP2-mediated angiogenesis and tumor vessel abnormalization.** SOX7 is a proangiogenic transcriptional factor. To verify this, SOX7 expression was reduced in HUVECs by a lentivirus expressing shRNA (Supplementary Fig. 6a). SOX7-knockdown impaired endothelial cell proliferation, migration, and tube formation (Supplementary Fig. 6b-d); and this finding was consistent with that observed in SHP2-knockdown cells.

**Fig. 3 SHP2 inhibitor reduces tumor angiogenesis and growth. a** Tumors were established in BALB/c nude mice by subcutaneous injection of LM3 cells expressing SHP2$^{WT}$ or SHP099-resistant mutant (SHP2$^{T253M/Q257L}$, SHP2_TM/QL) and treated with vehicle, SHP099 (75/38.7/18.75 mg/kg body weight, daily) or cediranib (1.5 mg/kg body weight, daily) from day 3. **b** Images for LM3 tumors. Scale bar: 10 mm. **c, d** The volumes (**c**) and weights (**d**) for LM3 tumors. Tumor weights were measured 13 days after cancer cell injection. Quantitative data were shown as mean ± SEM. *p*-values were shown and generated by two-way ANOVA with Tukey's post hoc test or by one-way ANOVA with multi-comparisons. *n* = 5 for SHP2$^{WT}$ + control group, *n* = 7 for the other each group. **e–h** Immunofluorescence staining for CD31 (**e, f**), αSMA (**g**), Collagen IV (**h**, Col4) in LM3 tumors. Scale bar: 100 μm. *n* = 5 for SHP2$^{WT}$ + control group, *n* = 7 for the other each group. **i–l** Vessel density (**i**), branching index (**j**), pericyte coverage (**k**), basement membrane coverage (**l**) were measured by using Image J software and quantitative data were shown as mean ± SEM. *p*-values were shown and generated by one-way ANOVA with multi-comparisons. *n* = 5 for SHP2$^{WT}$ + control group, *n* = 7 for the other each group. Source data are provided as a Source data file.

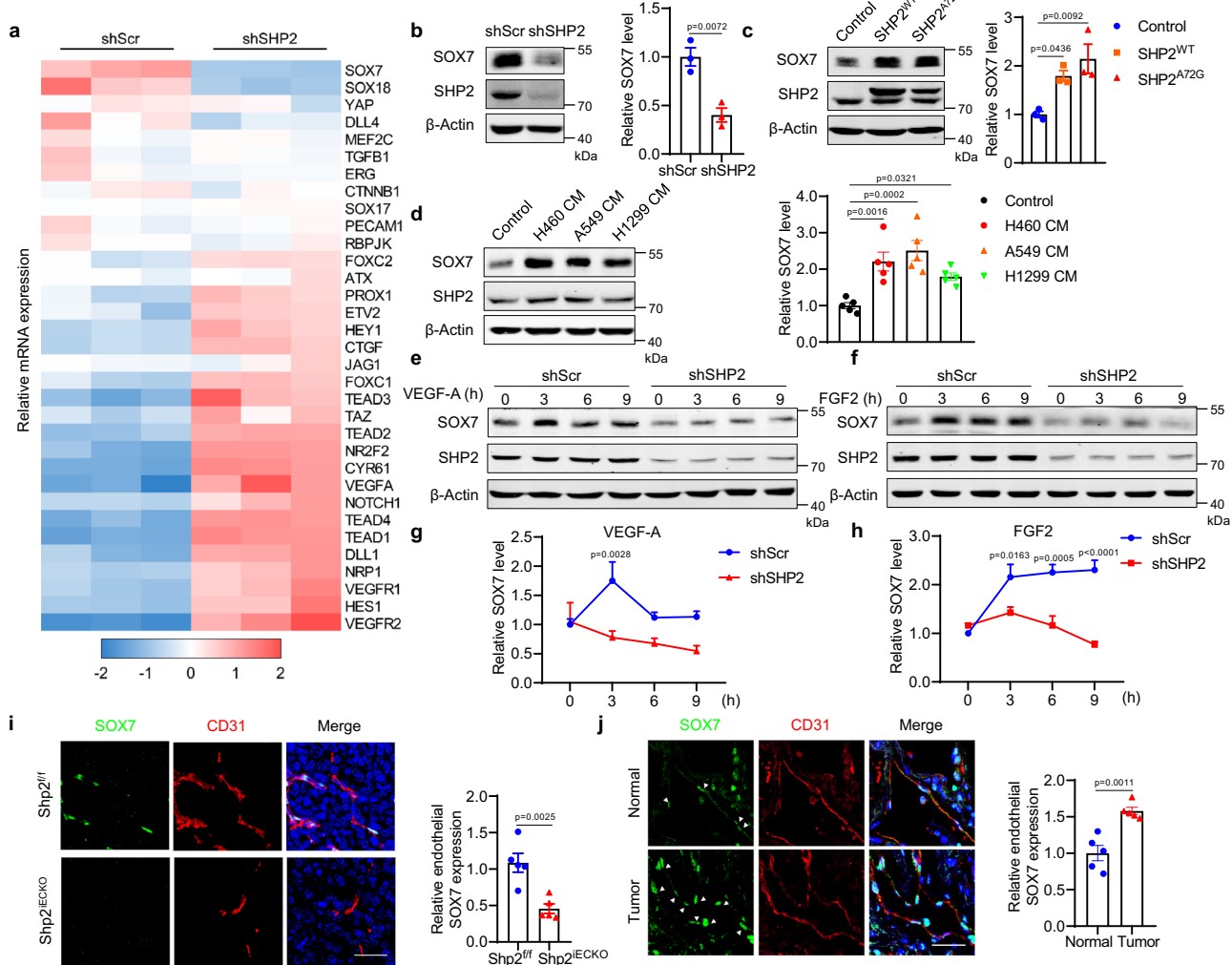

**Fig. 4 SHP2 regulates SOX7 expression. a** Heatmap showing qPCR results of angiogenesis-related genes in SHP2 knockdown (shSHP2) and control (shScr) HUVECs. **b** Western blot for SOX7 expression in SHP2-knockdown and control HUVECs. β-Actin was used as a loading control. Quantitative data were shown as mean ± SEM for three independent experiments. *p*-value was shown and generated by using the two-tailed Student's *t*-test. **c** Western blot for SOX7 in overexpression of SHP2 or its A72G active mutation and control HUVECs. β-Actin was used as a loading control. Quantitative data were shown as mean ± SEM for three independent experiments. *p*-values were shown and generated by one-way ANOVA with multi-comparisons. **d** Western blot for SOX7 in HUVECs treated with various NSCLC cancer cell-conditioned media for 24 h. β-Actin was used as a loading control. Quantitative data were shown as mean ± SEM for five independent experiments. *p*-values were shown and generated by one-way ANOVA with multi-comparisons. **e–h** Western blot for SOX7 expression in SHP2 knockdown and control HUVECs treated with VEGF-A (10 ng/ml) or FGF2 (10 ng/ml) for indicated time. β-Actin was used as a loading control. Quantitative data were shown as mean ± SEM for three independent experiments. *p*-values were shown and generated by two-way ANOVA with Bonferroni's multiple comparisons test. **i** Representative images showing SOX7 expression (green) in CD31$^+$ vessels (red) in LLC tumors from Shp2$^{f/f}$ (*n* = 5) and Shp2$^{iECKO}$ (*n* = 5) mice. Endothelial SOX7 was analyzed by the Image J software and quantitative data were shown as mean ± SEM. *p*-value was shown and generated by using the two-tailed Student's *t*-test. Scale bar: 100 μm. **j** Representative images showing SOX7 expression (green) in CD31$^+$ vessels (red) in NSCLC tumor tissues and paired adjacent normal tissues (*n* = 5). SOX7 expression in endothelial cells was analyzed by the Image J software and quantitative data were shown as mean ± SEM. *p*-value was shown and generated by using the two-tailed Student's *t*-test. Scale bar: 100 μm. Source data are provided as a Source data file.

Notably, SOX7 re-expression restored the endothelial function in SOX7-knockdown cells (Supplementary Fig. 6b-d), indicating that shRNA-mediated SOX7 knockdown was specific. Next, SOX7 was re-expressed in SHP2-knockdown HUVECs (Supplementary Fig. 6e), and the defects in cell proliferation, migration, and tube formation were eliminated by SOX7 re-expression (Supplementary Fig. 6f-h). SOX7 overexpression in HUVECs did not affect endothelial cell migration and tube formation, but increased cell proliferation. Finally, an AAV vector was constructed to express SOX7 in response to Cre recombinase (Supplementary Fig. 6i). SOX7 re-expression in tumor endothelial cells restored tumor growth and angiogenesis in Shp2$^{iECKO}$ mice (Fig. 5a–c). SOX7 and ZsGreen immunofluorescence staining demonstrated that SOX7 was effectively and specifically induced in the tumor endothelial cells of Shp2$^{iECKO}$ mice (Supplementary Fig. 6j, k). Importantly, the decrease in tumor necrosis and hypoxia in Shp2$^{iECKO}$ mice was significantly reversed by the expression of SOX7 (Fig. 5d, e). Furthermore, SOX7 expression in Shp2$^{iECKO}$ mice increased the vessel density and branching while decreasing pericyte and BM coverage; that is, SOX7 expression restored vascular abnormalization (Fig. 5f–m). The successful rescue experiments demonstrated that SOX7 is the key downstream mediator of SHP2 in regulating angiogenesis and vascular abnormalization in tumors.

**SHP2 regulates SOX7 expression through c-Jun signaling**. To elucidate the link between SHP2 and the transcriptional regulation of SOX7, the candidate transcription factors for SOX7 were explored in three different transcriptional factor databases. Four factors were presented in all three databases; among the factors, c-Jun belongs to the MAPK pathway regulated by SHP2 (Supplementary Fig. 7a). The activation of ERK, p38, and c-Jun was analyzed in SHP2-knockdown HUVECs. ERK and p38 failed to respond to SHP2 knockdown in HUVECs; however, c-Jun was significantly deactivated upon SHP2-knockdown, as indicated by the utilization of two phosphorylation-specific antibodies (Fig. 6a and Supplementary Fig. 7b). The same reduction pattern of c-Jun activation was observed in Shp2$^{iECKO}$ mouse lung endothelial cells (Fig. 6b). The inhibition of SHP2 by RMC4550 in endothelial cells decreased the activation of c-Jun and ERK (Supplementary Fig. 7c). Moreover, immunofluorescence staining indicated that p-c-Jun significantly decreased in SHP2-knockdown and RMC4500-treated HUVECs (Supplementary Fig. 7d). c-Jun was activated by VEGF-A and FGF2 in endothelial cells, which was blocked by SHP2-knockdown (Fig. 6c). The plugs in the Shp2$^{iECKO}$ and control mice indicated a marked reduction in c-Jun activation upon Shp2 deletion (Supplementary Fig. 7e). Endothelial cells in the tumors of Shp2$^{iECKO}$ mice also exhibited reduced phosphorylated c-Jun (Fig. 6d). Next, c-Jun was overexpressed in HEK293 cells, which increased the activity of the SOX7 gene promoter and SOX7 expression (Fig. 6e, f). The JNK inhibitor SP600125 was used to inhibit c-Jun activation, which resulted in a significant reduction of SOX7 in HUVECs (Fig. 6g). Moreover, similar to SHP2 and SOX7, the conditioned media obtained from culturing cancer cells induced c-Jun expression and activation (Fig. 6h). Finally, the GSE118904 database indicated that the c-Jun and JunD levels in tumor endothelial cells were higher than those of normal endothelial cells (Supplementary Fig. 7f, g). Therefore, the regulation of SOX7 expression by SHP2 is mediated by c-Jun signaling.

**SHP2 stabilizes ASK1 to promote c-Jun signaling**. ASK1, an upstream factor in c-Jun signaling, is reportedly dephosphorylated and stabilized by SHP2[44]. In endothelial cells, SHP2 knockdown or inhibition decreased ASK1 at the protein level, but not at the mRNA

level (Fig. 7a and Supplementary Fig. 8a, b). The re-expression of SHP2 or its activated mutant (A72G), but not its catalytic-dead mutant (C495S), in SHP2-knockdown endothelial cells restored the protein expression of ASK1 (Fig. 7b). Moreover, the SHP2 inhibitor SHP099 inhibited ASK1 expression in HUVECs with the re-expression of wild-type SHP2, but not inhibitor-resistant mutant (T253M/Q257L) (Supplementary Fig. 8c), indicating that phosphatase activity was required for SHP2-mediated SOX7 expression. ASK1 was induced by VEGF-A or FGF2, which were blocked by SHP2-knockdown (Fig. 7c, d). Moreover, co-immunoprecipitation experiments showed that SHP2 interacted with ASK1 through its PTP domain, suggesting that ASK1 is a substrate for SHP2 (Fig. 7e). Furthermore, the ubiquitin conjugation assay was performed to explore whether SHP2 regulates ASK1 protein expression through inhibiting its ubiquitination degradation, and the results showed that SHP2-knockdown led to increased ubiquitin conjugation to ASK1 (Fig. 7f). Similar to that with SHP2-knockdown, a mutant that could mimic phosphorylated ASK1, i.e., ASK1$^{Y718E}$, increased the conjugation of ubiquitin (Fig. 7f). Furthermore, ASK1 expression was reduced in HUVECs by a lentivirus expressing shRNA, and SOX7 was decreased in ASK1-knockdown endothelial cells (Supplementary Fig. 8d, e). Similarly, the ASK1 inhibitor GS-4997 inhibited c-Jun activation and SOX7 expression in endothelial cells (Supplementary Fig. 8f). Finally, the level of ASK1 in tumor endothelial cells and plugs in Shp2$^{iECKO}$ mice was lower than that in the controls (Fig. 7g and Supplementary Fig. 8g). ASK1 expression was increased in endothelial cells treated with conditioned media of cancer cells (Fig. 7h), as well as the tumor endothelial cells of NSCLC tissues, similar to SHP2 (Fig. 7i). Together, SHP2 interacts with and stabilizes ASK1, which activates ASK1-c-Jun signaling to increase SOX7 expression in tumor endothelial cells.

**Targeting endothelial SHP2 inhibits tumor growth**. Finally, we explored whether Shp2 deletion in the endothelial cells of well-established tumors had anti-tumor effects. Tamoxifen was injected to delete Shp2 when the tumors reached a diameter of 5 mm (11 d after tumor cell injection, Fig. 8a). Similar to the results shown in Fig. 1, the deletion of Shp2 before the injection of cancer cells inhibited tumor growth (early deletion, Fig. 8b–d). Notably, the deletion of Shp2 after the tumors grew to 5 mm also significantly inhibited tumor growth, as indicated by the measured tumor volumes and weights (late deletion, Fig. 8b–d). These results support the targeting of endothelial SHP2 for anti-tumor therapy in cancers.

## Discussion

Protein tyrosine phosphatase SHP2 is important for vascular development and homeostasis. Gene deletion in vascular smooth muscle and endothelial cells results in embryonic lethality due to the occurrence of severe defects in vascular development[34,45]. In endothelial cells, SHP2 interacts with VEGFR and VE-cadherin, which are essential for angiogenesis and endothelial barrier function[27,31–33]. Here, we provide further convincing evidence to show that SHP2 regulates ASK1-c-Jun signaling, which controls proangiogenic SOX7 expression and pathological angiogenesis (Fig. 8e). Together with two recent studies[40,41], in which tumor angiogenesis was disrupted by SHP2 inhibitors, our results from a genetic mouse model strongly suggest the anti-angiogenic potency of SHP2 inhibitors in cancer therapies.

We determined that the mechanism by which SHP2 regulated SOX7 expression involved ASK1-c-Jun signaling (Fig. 8e). It should be emphasized that SOX7 re-expression rescued both the SHP2-inefficiency-induced endothelial dysfunction in vitro and SHP2-deletion-induced impairment of tumor angiogenesis in vivo, demonstrating the unique and critical role of this

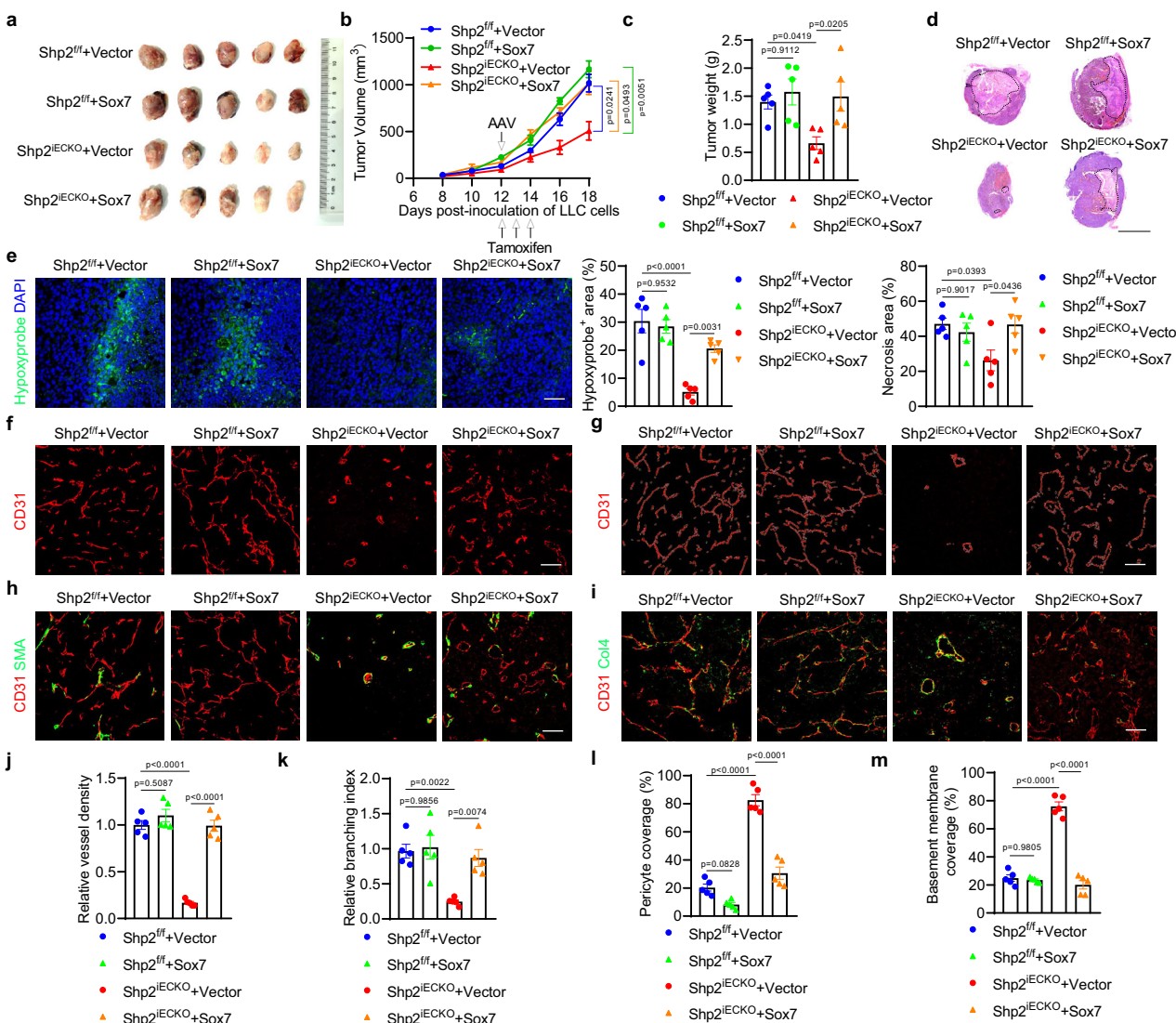

**Fig. 5 SOX7 is required for SHP2 to promote tumor angiogenesis and vessel abnormalization. a** Images of LLC tumors from Shp2[f/f] ($n = 5$) and Shp2[iECKO] ($n = 5$) mice treated with AAV-Sox7. **b, c** Tumor volumes (**b**) and tumor weights (**c**) of LLC tumors from Shp2[f/f] ($n = 5$) and Shp2[iECKO] ($n = 5$) mice treated with AAV-Sox7. Arrows indicated the time points for AAV and tamoxifen injection. Tumor weights were measured 18 days after cancer cell injection. Quantitative data were shown as mean ± SEM. $p$-values were shown and generated by two-way ANOVA with Tukey's post hoc test (**b**) or one-way ANOVA with multi-comparisons (**c**). **d** Representative images of LLC tumor necrosis. Necrosis areas were labeled by dot lines. Images were analyzed by the Image J software and quantitative data were shown as mean ± SEM. $n = 5$ in each group. $p$-values were shown and generated by one-way ANOVA with multi-comparisons. Scale bar: 10 mm. **e** Representative images of hypoxyprobe-1 labeled areas in LLC tumors. Images were analyzed by the Image J software and quantitative data were shown as mean ± SEM. $n = 5$ in each group. $p$-values were shown and generated by one-way ANOVA with multi-comparisons. Scale bar: 100 μm. **f–m** Immunofluorescence images of CD31 (**f, g**), αSMA (**h**), and collagen IV (**i**, Col4) in LLC tumors. Vessel density (**j**), branching index (**k**), pericyte coverage (**l**), basement membrane coverage (**m**) were measured by using the Image J software and quantitative data were shown as mean ± SEM. $n = 5$ in each groups. $p$-values were shown and generated by one-way ANOVA with multi-comparisons. Scale bar: 100 μm. Source data are provided as a Source data file.

proangiogenic factor in mediating the regulation of tumor angiogenesis by SHP2. The functions of SOX7 and other SOXF members in endothelial cells have been well studied. RNA-seq revealed that SOX7 was induced in endothelial cells by hypoxia and then regulates hypoxia-induced angiogenesis[46]. It has been previously established that SOX7 deletion in mouse endothelium results in embryonic lethality due to severe defects in the development of vasculature[47]. SOX7 and SOX17 are similar in both expression and function. SOX17 mediates Notch and Wnt signaling in the regulation of angiogenesis[48,49]. Loss-of-function and gain-of-function studies have demonstrated that SOX17 in endothelial cells promotes tumor angiogenesis and the formation

of vessel abnormalities[50]. A recent study reported that SOX17 is a regulator of coronary arteriogenesis[51]. Furthermore, studies using genetic mouse models revealed redundancies in the regulation of angiogenesis between SOX17 and SOX18[52]. In addition, SOX18 has been suggested to be essential in endothelial progenitor differentiation and tumor angiogenesis[53,54]. These findings emphasize the importance of the SOXF family in regulating vascular development and angiogenesis, including angiogenesis in tumors. Furthermore, we found that the ERK and p-38-MAPK pathways in endothelial cells were not regulated through SHP2; however, we observed that JNK-c-Jun signaling regulated the transcription of SOX7, and was controlled by SHP2. Finally, it has

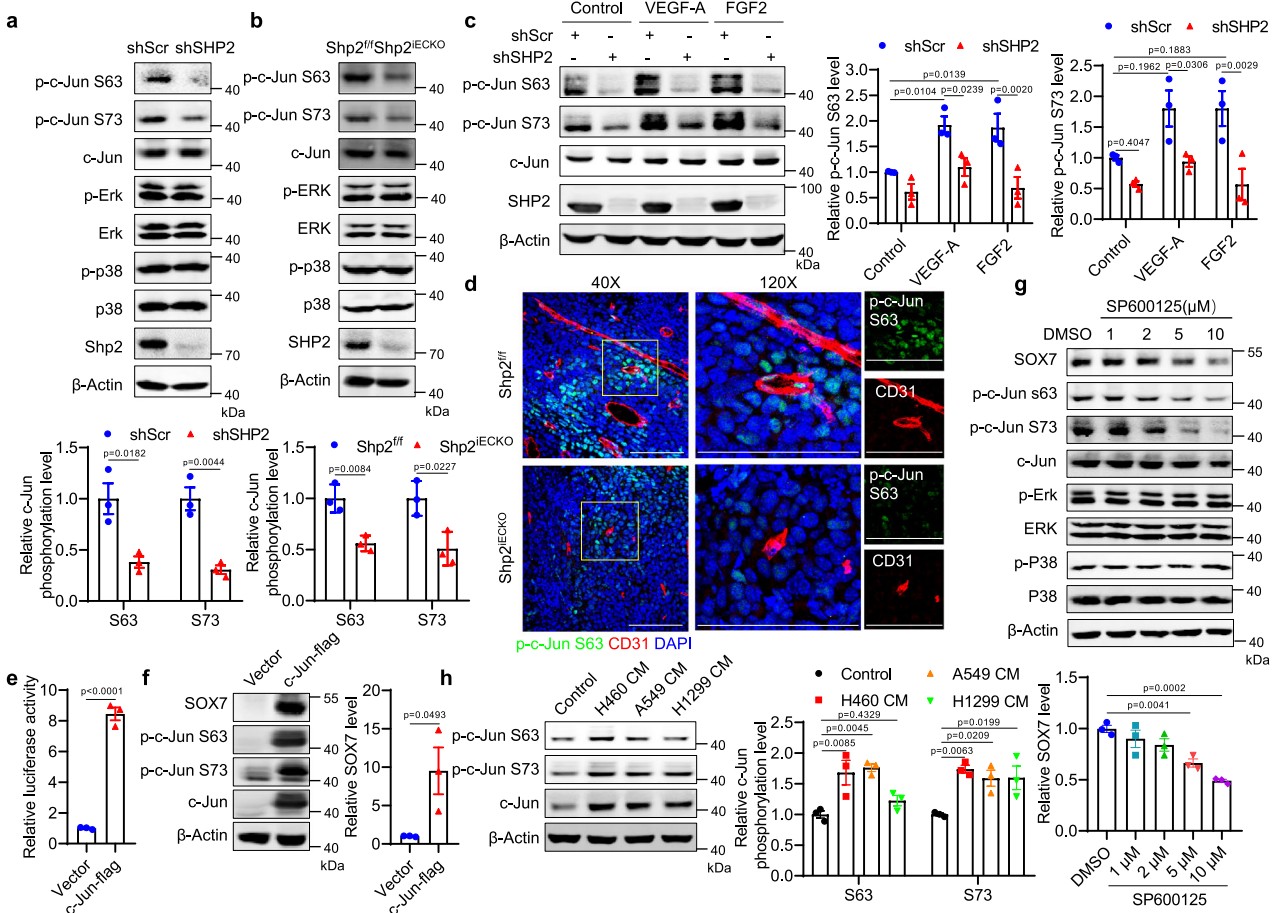

**Fig. 6 SHP2 regulates SOX7 expression through c-Jun signaling. a, b** Western blot for c-Jun, ERK, and P38 and associated phosphorylated forms in SHP2-knockdown HUVECs and MLECs isolated from Shp2[f/f] and Shp2[iECKO] mice. β-Actin was used as a loading control. Quantitative data were shown as mean ± SEM for three independent experiments. *p*-values were shown and generated by one-way ANOVA with multi-comparisons or the two-tailed Student's *t*-test. **c** Western blot for c-Jun and associated phosphorylated forms in SHP2 knockdown and control HUVECs treated with VEGF-A (10 ng/ml) or FGF2 (10 ng/ml) for 10 min. β-Actin was used as a loading control. Quantitative data were shown as mean ± SEM for three independent experiments. *p*-values were shown and generated by two-way ANOVA with Bonferroni's multiple comparisons test. **d** Representative images showing p-c-Jun S63 level (green) in CD31[+] vessels (red) in LLC tumors in Shp2[f/f] (*n* = 5) and Shp2[iECKO] (*n* = 5) mice. Scale bar: 50 μm; inset 100 μm. **e** Luciferase reporter assay in HEK293 cells with c-Jun overexpression. The data were normalized to the Renilla luciferase activity. Quantitative data were shown as mean ± SEM for three independent experiments. *p*-value was shown and generated by the two-tailed Student's *t*-test. **f, g** Western blot for SOX7, c-Jun, and its phospho-forms in HEK293 cells with c-Jun overexpression (**f**) and SOX7, c-Jun, phosphor-c-Jun, ERK, phosphor-ERK, P38, and p-P38 in HUVECs treated with various concentrations of JNK inhibitor SP600125 (24 h) (**g**). β-Actin was used as a loading control. Quantitative data were shown as mean ± SEM for three independent experiments. *p*-values were shown and generated by using the two-tailed Student's *t*-test or by one-way ANOVA with multi-comparisons. **h** Western blot for c-Jun and associated phosphorylated forms in HUVECs treated with various NSCLC cancer cell-conditioned media for 24 h. β-Actin was used as a loading control. Quantitative data were shown as mean ± SEM for three independent experiments. *p*-values were shown and generated by one-way ANOVA with multi-comparisons. Source data are provided as a Source data file.

been reported that SHP2 dephosphorylates and stabilizes ASK1 to stimulate downstream JNK-c-Jun signaling[44]. Therefore, we proposed proangiogenic SHP2-ASK1-c-Jun-SOX7 signaling pathway responsible for regulating tumor angiogenesis and vessel abnormalization. ASK1, a MAP kinase, regulates JNK-c-Jun activation and the apoptosis of endothelial cells in response to extracellular stimuli, including TNFα, blood flow, and intracellular redox status[55–58]. It has been asserted that ASK1-interacting protein (AIP1) regulates angiogenesis and lymphangiogenesis by regulating the stability of VEGFR3[59]. Both gene deletion and the pharmacological inhibition of ASK1 in tumor endothelial cells prevented tumor growth, and the mechanism relies on increasing the integrity of the endothelial barrier and decreasing macrophage infiltration[60]. Previous studies have not analyzed tumor angiogenesis and vessel normalization, although it has been

determined that VE-cadherin, an important endothelial junction molecule involved in angiogenesis, is protected by ASK1. We found that the expression levels of all components of the SHP2-ASK1-c-Jun-SOX7 signaling axis were upregulated or activated in tumor-associated endothelial cells, highlighting the importance of this signaling axis in controlling tumor angiogenesis and vessel abnormality. As a limitation, the mechanism by which SHP2 is activated in tumor-associated endothelial cells was not explored in this study.

As allosteric inhibitors for SHP2 have already been developed, their anti-tumor activity in cancer cells has been investigated extensively[12,14–16,18]. In addition to the cancer cell itself, the tumor microenvironment is important for tumor growth and malignant progression, including metastasis and drug resistance. Macrophages are important components of the tumor microenvironment.

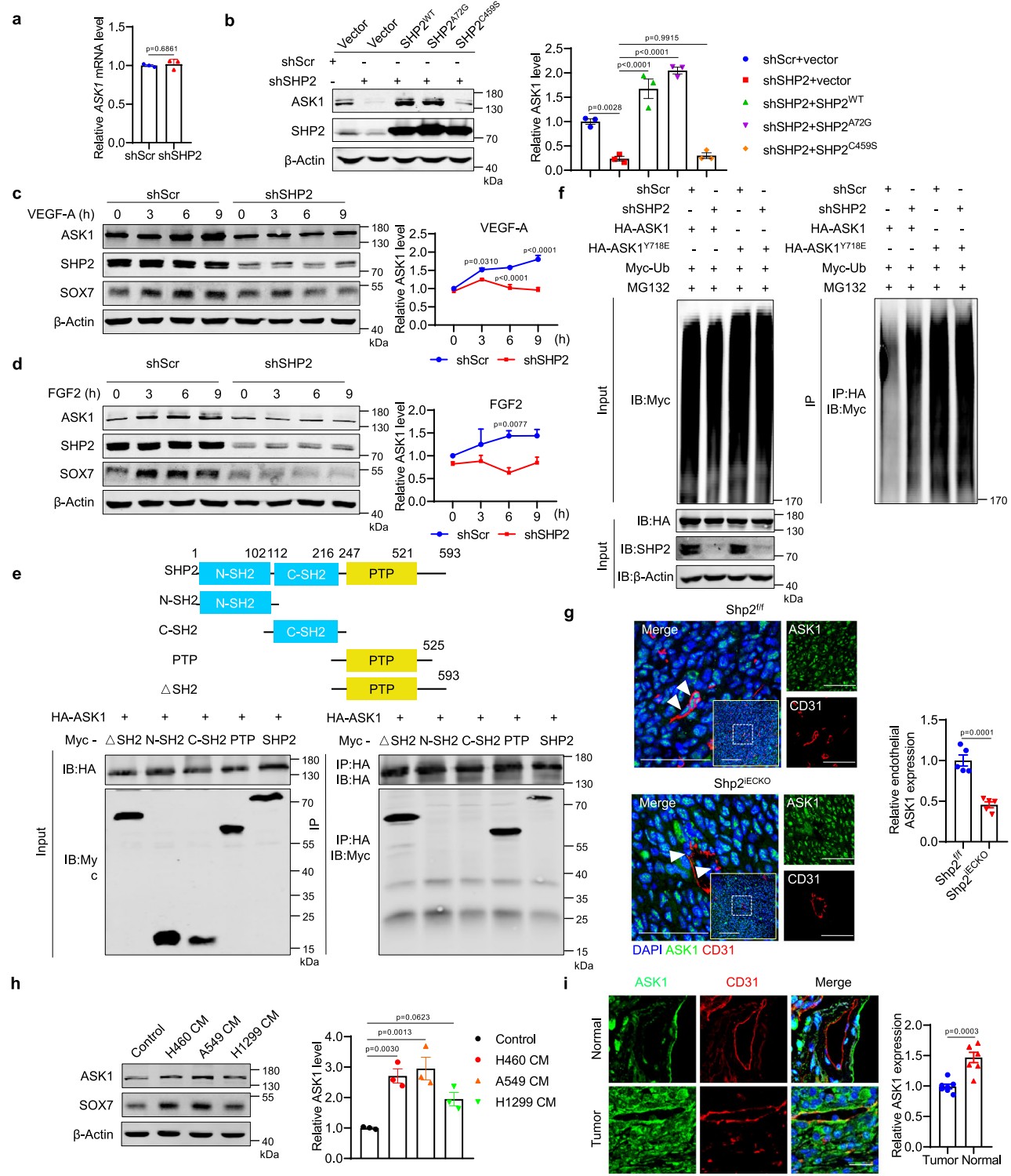

We have demonstrated that macrophages with SHP2 deletion switched to M2 polarization, which has been suggested to favor tumor growth[61]. However, our research showed that the deletion of Shp2 in macrophages suppressed tumor growth by promoting the interaction between macrophages and T cells in tumor microenvironments[21]. A recent finding demonstrated that SHP2 inhibition in tumor cells increases the expression of CXCR2 ligands, which suppresses M2 macrophages and promotes gMDSC infiltration, suggesting combined inhibition of SHP2 and CXCR2 in lung cancer[62]. In addition, SHP2 is a positive regulator of TGFβ-

Smad signaling[45,63–65], which induces the differentiation of fibroblasts into myofibroblasts; however, the role of SHP2 in tumor-associated fibroblasts remains unknown. Moreover, SHP2 mediates immunoinhibitory receptor signaling, including PD-1 and cytotoxic T lymphocyte-associated protein 4 (CTLA-4)[66]; accordingly, it has been found that SHP2 deletion in T cells promotes tumor growth and metastasis[67]. In this study, we found that the gene ablation of Shp2 in tumor endothelial cells impaired tumor growth and angiogenesis while promoting the normalization of tumor vasculature. Two recent studies used SHP2 inhibitors to demonstrate

**Fig. 7 SHP2 positively regulates c-Jun/SOX7 signaling by inhibiting ASK1 degradation. a** qPCR for *ASK1* in SHP2-knockdown HUVECs. Quantitative data were shown as mean ± SEM for three independent experiments. p-value was shown and generated by using the two-tailed Student's *t*-test. **b** Western blot for ASK1 in SHP2-knockdown HUVECs with re-expression of SHP2WT or its activated (A72G) and catalytic-dead (C459S) mutants. β-Actin was used as a loading control. Quantitative data were shown as mean ± SEM for three independent experiments. p-values were shown and generated by one-way ANOVA with multi-comparisons. **c, d** Western blot for ASK1 in HUVECs treated with VEGF-A (**d**; 10 ng/ml) or FGF2 (**e**; 10 ng/ml). β-Actin was used as a loading control. Quantitative data were shown as mean ± SEM for three independent experiments. p-values were shown and generated by two-way ANOVA with Tukey's post hoc test. **e** Western blot showing co-immunoprecipition assay for HA-tagged ASK1 and Myc-tagged SHP2 and its truncated mutations in HEK293 cells. Results were repeated for three independent experiments. **f** Ubiquitin conjugation assay for ASK1 and its Y718E mutation in HEK293 cells. Results were repeated for three independent experiments. **g** Representative images showing ASK1 expression (green) in CD31+ vessels (red) in LLC tumors in Shp2f/f (n = 5) and Shp2iECKO (n = 5) mice. Endothelial ASK1 were analyzed by the Image J software and quantitative data were shown as mean ± SEM. Scale bar: Scale bars: 50 μm; inset 100 μm. p-value was shown and generated by using the two-tailed Student's *t*-test. **h** Western blot for ASK1 in HUVECs treated with various NSCLC cancer cell-conditioned media for 24 h. β-Actin was used as a loading control. Quantitative data were shown as mean ± SEM for three independent experiments. p-values were shown and generated by one-way ANOVA with multi-comparisons. **i** Representative images showing ASK1 expression (green) in CD31+ vessels (red) in NSCLC tissues (n = 7) and paired adjacent normal tissues (n = 7). Quantitative data were shown as mean ± SEM. p-value was shown and generated by using the two-tailed Student's *t*-test. Scale bar: 10 μm. Source data are provided as a Source data file.

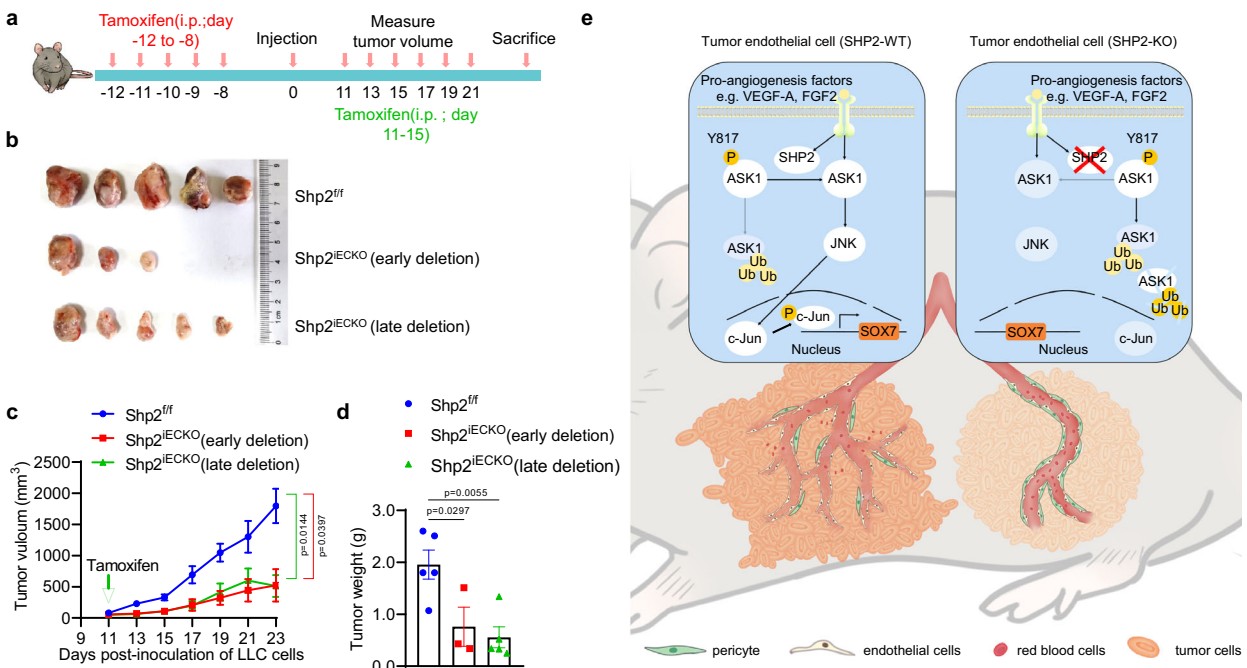

**Fig. 8 Targeting endothelial SHP2 inhibits tumor growth. a** Schematic representation of the experimental model to delete Shp2 after tumor growth (late deletion). For late Shp2 deletion, tumors were grown in Shp2f/f and Shp2iECKO mice before tamoxifen was administered when the tumors reached a diameter of 5 mm. **b** Images of explant LLC tumors from Shp2f/f and Shp2iECKO mice (early deletion or late deletion). Scale bar: 10 mm. **c, d** The volumes (**c**) and weights (**d**) of LLC tumors from Shp2f/f (n = 5) and Shp2iECKO (early deletion, n = 3; late deletion, n = 5) mice. The weights were recorded 11 days after cancer cell injection. Quantitative data were shown as mean ± SEM. p-values were shown and generated by two-way ANOVA with Tukey's post hoc test (**c**) or by using the two-tailed Student's *t*-test (**d**). **e** Schematic illustration showing the role and mechanism of SHP2 in regulating tumor angiogenesis and vessel abnormality. In tumor-associated endothelial cells, expression of protein tyrosine phosphatase SHP2 is upregulated in response to signals from cancer cells. SHP2 dephosphorylates and stabilizes ASK1, which promotes c-Jun signaling to induce SOX7 expression. SOX7 is a proangiogenic factor and promotes tumor angiogenesis to produce tumor vessels with inefficiency in pericyte coverage, barrier integrity, and vessel perfusion. When SHP2 is genetically deleted or pharmacologically inhibited, through ASK1-c-Jun signaling, SOX7 expression is reduced. Reduced SOX7 expression results in a decrease in tumor angiogenesis while produces tumor vessels with well pericyte coverage, barrier integrity, and vessel perfusion. SHP2-ASK1-c-Jun-SOX7 is an important signaling axis in regulation tumor angiogenesis and vessel normalization, thus rendering the signaling pathway, especially SHP2, whose allosteric inhibitor is available, a valuable target for the development of anti-angiogenic therapy in cancers. Source data are provided as a Source data file.

their proangiogenic function in mouse tumor models using SHP2-independent cancer cells or cancer cells with SHP2 genetic deletion[40,41]. SHP2 was ubiquitously expressed; however, the pharmacological inhibition of SHP2 appeared to be safe for normal tissues[68]. Together, SHP2 inhibition exhibited great tumor cell-intrinsic and -extrinsic killing actions, which are more favorable to SHP2-independent tumors.

Using inducible and endothelial-cell-specific gene-deleted mice, we demonstrated that SHP2 in tumor-associated endothelial cells is activated to promote tumor growth, angiogenesis, and vascular abnormalization. The ASK1-c-Jun-SOX7 signaling axis, controlled by SHP2, is also activated and contributes to tumor angiogenesis. These findings highlight the value of SHP2 inhibitors in the development of anti-angiogenic cancer therapy.

## Methods

**Mice.** Shp2f/f mice were generous gifts from Dr. Gen-Sheng Feng (University of California, San Diego, USA). Shp2f/f mice were crossed with Cdh5-CreERT2 mice to generate Shp2f/f: Cdh5-CreERT2 (Shp2iECKO, C57BL/6) mice. Shp2iECKO mice were intraperitoneally injected with tamoxifen (20 mg/kg, i.p.) for 5 d to delete Shp2. Shp2f/f mice with the same tamoxifen injection were used as controls. Genotyping was performed by PCR of DNA samples using the following primers: Shp2f/f forward, ACGTCATGATCCGCTGTCAG; Shp2f/f reverse, ATGGGAG GGACAGTGCAGTG; Cdh5-CreERT2 forward, CCAAAATTTGCCTGCATTAC CGGTCGATGC; Cdh5-CreERT2 reverse, ATCCAGGTTACGGATATAGT. All mice were cultured in suitable temperature and humidity environment and fed with sufficient water and food (25 °C, suitable humidity (typically 50%), 12 h dark/light cycle). All animal protocols were approved by the Animal Care and Use Committee of the Zhejiang University School of Medicine.

**Cells.** HUVECs were isolated from umbilical cord vein by lavaging with 0.2% (w/v) collagenase solution at 37 °C for 15 min, the cells were collected and suspended in complete M199 medium. HUVECs between passages 4 and 6 were cultured in a complete medium containing 53% M199 (Corning Incorporated), 37% human endothelial serum-free medium (Thermo Fisher Scientific), and 15 μg/mL of endothelial cell growth supplement (Sigma-Aldrich). Human cerebral microvessel endothelial cells (hCMECs) were purchased from Zhejiang Meisen Cell Technology Co., Ltd. Primary mouse lung endothelial cells (MLECs) were isolated from 6- to 8-week-old mice. Briefly, 1 month after tamoxifen administration, mice were anesthetized with 80 mg/kg ketamine and 12 mg/kg xylazine, and then subjected to perfusion by intracardiac injection of PBS (0.1% BSA) for blood removal. Lung tissue was then dissected and cut into 1–2-mm pieces, and digested with a solution containing type-I collagenase (2 mg/mL, Biosharp), dispase (1 mg/mL, Roche), and DNase (10 μg/mL, Roche) in DPBS for 45 min at 37 °C in a rotatory shaker (at 80 rpm). After digestion, the enzymes were neutralized by DMEM plus 20% FBS. A single-cell suspension was prepared using a 40-μm cell strainer. Finally, the cells were further purified using anti-mouse CD31-conjugated magnetic beads (Invitrogen) and maintained in a complete medium containing DMEM with 10% FBS, endothelial cell growth supplement (15 μg/mL, Sigma-Aldrich), 1% nonessential amino acids (Gibco), and heparin (0.1 mg/mL, Solarbio).

Mouse Lewis lung cancer (LLC), B16 melanoma cells, and E0771 purchased from ATCC were cultured in DMEM. Non-small cell lung cancer (NSCLC) (H460, A549, H1299) and hepatocellular carcinoma LM3 cells purchased from ATCC were grown in RPMI-1640 (Thermo Fisher Scientific). The media were supplemented with 10% FBS, 100 IU/mL penicillin, and 100 μg/mL streptomycin, and the cells were incubated at 37 °C under 5% CO$_2$.

For the double luciferase reporter assay, the promoter fragment containing multiple predicted c-Jun binding sites (nucleotides -2000 to 100 of human SOX7 gene loci) was cloned into a pGL3-basic vector. In 12-well plates, HEK293 cells were transfected with vector or c-Jun for 24 h. The double luciferase reporter assay was conducted following the manual for the Dual-Luciferase reporter gene assay kit (Promega). The reporter gene activity was detected by GloMax® 20/20 Luminometer (Promega) and normalized to the activity of Renilla luciferase.

To construct SHP2WT and inhibitor-resistant mutant SHP2 (SHP2T253M/Q257L) LM3 cell lines, SHP2WT and SHP2T253M/Q257L cDNA were constructed into the PLVX-NEO vector and packaged into lentiviruses. LM3 cells were infected with shSHP2-3'utr lentivirus and screened by puromycin. The SHP2 knockdown LM3 cells were then infected with SHP2WT or SHP2T253M/Q257L lentivirus, and neomycin was further used to screen stable transgenic strains.

**Mouse tumor models.** To establish syngeneic mouse tumor models, suspensions of LLC ($4 \times 10^5$ cells in 100 μL) or B16 ($8 \times 10^5$ cells in 100 μL) cells were subcutaneously implanted in the dorsal flank of 8- to 10-week-old male mice. For the orthotopic mouse breast tumor model, E0771 cells ($8 \times 10^5$) were suspended in 100 μL of Matrigel (50% v/v; Corning) in RPMI1640 medium and injected into the mammary fat pads of 8-week-old female mice. The tumor volumes were measured using a digital caliper every 2 days and calculated according to the following formula: $V = 0.52 \times L \times W^2$ (V, tumor volume; L, longest diameter of the tumor; W, perpendicular diameter of L)[69,70]. Sixteen or eighteen days after cell injection, tumors were harvested and fixed with 4% paraformaldehyde (PFA) for further histological analysis.

**Matrigel plug assay.** Matrigel (500 μL), supplemented with the recombinant mouse FGF2 and VEGF-A (400 ng/mL each) and heparin (50 units/mL) was injected subcutaneously into the flanks of 8- to 10-week-old mice, and they were sacrificed 7 days after injection. The Matrigel plugs were removed, photographed, and fixed in 4% PFA for further analysis.

**Mouse aortic ring assay.** The thoracic aortas were removed from the mice under anesthesia and cut into 1-mm rings. The aortic rings were then placed between two layers of 100 μL growth factor-reduced Matrigel (Corning) supplemented with heparin (20 U/mL, Solarbio), VEGF-A (100 ng/mL, Novoprotein), and FGF2 (100 ng/mL, Novoprotein), and incubated in culture medium (10% FBS, 100 IU/ mL penicillin and 100 μg/mL streptomycin). The culture medium was changed

every other day. The sprouting area was recorded under a phase-contrast microscope on day 7 and measured using Image J 1.49 v.

**Co-immunoprecipitation, immunoblotting, and immunofluorescence staining.** For western blotting, the total protein concentrations (20 μg) from MLECs or HUVECs were separated using SDS-PAGE gels and transferred onto nitrocellulose membranes (Pall, Port Washington). The primary antibodies used are shown in Supplemental Table 1.

For co-immunoprecipitation, cells were harvested and lysed in a cell lysis buffer used for western blotting and IP (Beyotime), protease inhibitor cocktail (Roche), and PhosSTOP (Roche) on ice. Protein G Dynabeads were pretreated with the primary antibody for 10 min at 25 °C and then washed with PBST three times. The cell lysates were then incubated overnight with antibody-conjugated beads at 4 °C. The immunoprecipitates were then used for western blotting by odyssey (version 3.0.29) or LI-COR image studio (version 5.2).

For tissue immunofluorescence staining, samples were fixed in 4% PFA, dehydrated in a 30% sucrose solution for 24 h, and embedded using the Tissue-Tek OCT compound. Frozen blocks were cut into 10-μm-thick sections. For cell immunofluorescence staining, endothelial cells were fixed with 4% PFA and permeabilized with 0.5% Triton for 15 min. The samples were blocked with 5% goat or donkey serum in PBST and incubated overnight at 4 °C with primary antibodies (Supplemental Table 1). After performing multiple washes, the samples were incubated for 1 h at room temperature with the secondary antibodies (Supplemental Table 1). The nuclei were stained with 4′,6-diamidino-2-phenylindole (DAPI) (Beyotime). The samples were then mounted with fluoromount-G (Southern Biotech), and immunofluorescence images were acquired using a confocal microscope (FV3000, Olympus). The images were further processed using ImageJ 1.49v or OlyVIA VS200.

**Quantitative PCR.** The total RNA was isolated from HUVECs and MLECs by TRIzol and reverse-transcribed to cDNA using the ReverTraAce qPCR RT kit (Toyobo Inc.). Quantitative RT-PCR was conducted using SYBR green dye by CFX96 Touch Real-Time PCR Detection System (Bio-Rad) and the primers listed in Supplemental Table 2 and Supplementary Table 3. Gene expression was calculated using the equation RQ = $2^{-\triangle\triangle Ct}$ and normalized to GAPDH or 18S RNA values.

**Pimonidazole staining, vascular leakage, and perfusion assay.** To measure tumor hypoxia, the mice were injected with Hypoxyprobe-1 (60 mg/kg, Hypoxyprobe) i.p. 1 h before they were sacrificed. Tumors were then removed and embedded in Tissue-Tek OCT. FITC-conjugated mouse anti-pimonidazole monoclonal antibody was applied following the manufacturer's protocol. To evaluate vascular permeability, 100 μL of FITC-conjugated dextran (25 mg/mL, 70 kDa, Sigma-Aldrich) was intravenously injected and the mice were perfused with 1% PFA to remove circulating dextran after 30 min. Tumors were then dissected and embedded in the Tissue-Tek OCT compound for further analysis. To measure vessel perfusion, DyLight 488-conjugated Tomato lectin (1 mg/mL, 100 μL, Vector Laboratories) was i.v. injected 30 min before sacrifice, and the tumors were harvested for further analysis.

**EdU incorporation assay.** Cells were seeded in the 24-well plates at a density of $4 \times 10^4$ cells per well. After overnight incubation, the cells were incubated with EdU (Beyotime) for 5 h at 37 °C and then fixed in 4% PFA. After washing with PBS, the cells were treated with 0.5% Triton X-100 for 15 min at room temperature for permeabilization. The cells were incubated and protected from light using a click additive solution and stained with DAPI. Images were obtained using a confocal microscope (FV3000, Olympus).

**Transwell cell migration assay.** HUVECs or MLECs suspended in the complete medium were seeded in the upper chamber. Four hours later, the medium in the upper chamber was changed to 0.1% FBS in M199, and the lower chamber was filled with 600 μL of complete medium. After incubation for 24 h at 37 °C, non-migrating cells were removed and cells that migrated through the membrane were fixed in 4% PFA and stained with crystal violet solution (Beyotime), after photographed, the crystal violet-stained cells were incubated with 33% acetic acid, and absorbance was detected at 570 nm (SynergyMx M5, Molecular Devices).

**Endothelial cell tube formation.** Cells were seeded in Matrigel-coated u-slides at a density of $1 \times 10^4$ cells per well (Ibidi, Germany) and incubated at 37 °C for 4 h and for a further 15 min with calcein (Yeasen). Images were acquired using a fluorescence microscope (IX70, Olympus). Five different fields for each condition were quantified by counting the number of junctions, total tube length, and total branching length using ImageJ 1.49v.

**NSCLC tissues.** NSCLC and matched adjacent normal tissue specimens were collected from the First Affiliated Hospital, College of Medicine, Zhejiang University (Hangzhou, China), and confirmed by pathological diagnosis. This study was approved by the Research Ethics Committee of the First Affiliated Hospital,

College of Medicine, Zhejiang University and abided by the Declaration of Helsinki principles. Written informed consent was obtained from all patients prior to the study.

**Quantification and statistical analysis**. Statistical analysis was performed using Prism software (GraphPad Inc, version 6.0 and 8.0). All quantitative data are presented as the mean ± SEM. Unpaired Student's *t*-tests were used for comparisons between two groups. Multiple group comparisons were conducted by one-way or two-way ANOVA followed by Tukey's post hoc tests or multi-comparisons. $p < 0.05$ was considered statistically significant.

**Reporting summary**. Further information on research design is available in the Nature Research Reporting Summary linked to this article.

## Data availability
Figure 1d, Supplementary Fig. 7f, g were generated from the NCBI's Gene Expression Omnibus (GEO) database GSE118904[42]. The transcription factor prediction data were generated by Qiagen Transcript Discovery plugin and from PROMO (http://alggen.lsi.upc.es/cgi-bin/promo_v3/promo/promoinit.cgi?dirDB=TF_8.3) and JASPER (http://jaspar.genereg.net/). The remaining data supporting the findings of this study are available within the article, Supplementary Information, or source data file. Source data are provided with this paper.

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

## Acknowledgements

This work was supported by the Key Research and Development Project of the Ministry of Science and Technology of China (2016YFA0501800 to Y.K.); the National Natural Science Foundation of China (32070952 and 31871399 to H.C., 81873418 to Y.K., and 32000799 to J.Z.), and the Zhejiang Provincial Natural Science Foundation of China (LZ18H020001 to H.C.). We thank Shuangshuang Liu and Qiong Huang from the core facility platform of Zhejiang University School of Medicine for providing technical support. We would like to thank *Editage* for English language editing.

## Author contributions

Z.X., H.C., and Y.K. designed and analyzed the experiments and wrote the manuscript. X.Z., Z.X., C.G., Q.Y., Yue S., Yi.S., J.Z., J.H., C.Z., and Y.H. performed the experiments. C.G. and Z.X. illustrated the working model and edited the manuscript.

## Competing interests
The authors declare no competing interests.
