## [Peer Review File · Nature Communications]

Endothelial deletion of SHP2 suppresses tumor angiogenesis and promotes vascular normalizationREVIEWER COMMENTS

Reviewer #1 (Remarks to the Author):

The manuscript by Xu et al describes a novel and very unexpected anti cancer effect of SHP2/PTPN11 inhibitors. These drugs are considered to have anti cancer effects by blocking signals from receptor tyrosine kinases (RTKs) to downstream effectors. As such, most trials with these drugs are focussing on RTK mutant cancers. The present manuscript demonstrates that SHP2 also has a critical role in the endothelium of the cancer, as they demonstrate that deletion of SHP2 in only endothelial cells has a major effect on tumor growth. As such, these findings shed a new light on how SHP2 inhibitors can inhibit cancer growth. This finding will without doubt be of interest to a wide readership.

The biological data are supported by mechanistic studies that point towards a major role for SOX7 downstream of SHP2 in endothelial cell function.

There are a few points that the authors should address:

1. The manuscript should be corrected by a person that is fluent in English.
2. Figure 1h needs a legend for the staining
3. Figure 3a: why focusing only on downregulated genes? The upregulation of VEGFR is also quite strong and interesting, but it is not even discussed.
4. Figure 4: How was the AAV vector delivered? I can guess intranasal or intratracheal injection, but it is not mentioned
5. The rationale behind looking at cJun as SOX7 regulator is unclear.

Reviewer #2 (Remarks to the Author):

In this manuscript, Zhang et al. investigated the role of SHP2 in tumor-associated angiogenesis. The authors concluded that tumor-associated endothelial cells have increased levels of total and activated SHP2 that is required to de-phosphorylate and stabilize ASK1, activate c-Jun and consequently increase the SOX7 transcription. SHP2 role in endothelial cell functions has been extensively studied and SHP2 has been already implicated in regulation of vessel stability and permeability via different molecular mechanisms. Also, it was already shown that inhibition of SHP-2 suppresses angiogenesis in vitro and in vivo. The noteworthy result from the current manuscript is the new molecular mechanism implicating SHP2 in increasing the expression of SOX7 in endothelial cells. Even though the mechanism is somewhat experimentally defined, mostly in vitro, additional evidence needed and specified below. Also, it is unclear how the new mechanism is related to the previously shown role of SHP2: the authors and others have previously shown that SHP2 depletion caused endothelial cell junction disruption and lung edema. Clinical relevance of the conclusions is weak and needs to be clarified: is it relevant to the specific types of cancer or ubiquitous to all cancers.

Major points are outlined below.

Major concerns:

1. Since endothelial cells in different organs are phenotypically and functionally different, the authors need to use orthotopic tumor models to assess tumor angiogenesis, which is also tissue specific. They need to inoculate LLC tumor cells into the lung tissue and B16 tumor cells intradermally to make the clinically relevant models of these two types of cancer.
2. In their previous publication, the authors demonstrate that conditional deletion of Shp2 in endothelial cells (EC) caused embryonic lethality associated with disruption of endothelial cell junctions and massive hemorrhage, and the inducible deletion of Shp2 in EC (Shp2iECKO) in adult mice resulted in lung vessel permeability. It is essential to measure lung vessel permeability in Shp2iECKO mice that are used for the experiments in this manuscript and their survival. Structural and functional changes in lung endothelial cells should be assessed at different time points after tamoxifen treatment, including at the time of tumor harvest, and whether those changes are similar or different in lung vs subcutaneous endothelial cells.
3. The conclusion that SHP2 is increased and activated in tumor-associated endothelial cells in vivo is not based on strong evidence. The immunofluorescent images of mouse, and especially human lung cancer tissues (Fig. 1b-c) are of low quality and must be replaced. High and low

magnifications of IF staining should be added. It should be also clarified how the relative p-SHP2 or SHP2 were quantified using these images. In addition, in it is unclear why the authors use lung EC to show protein levels of SHP2 (Fig. 1b-c), but mammary EC to show mRNA levels of SHP2 (Fig. 1d).

4. The statement that there are more α -SMA-positive endothelial cells in Shp2iECKO vessels does not make any sense since there are no such cells.

5. The statement that Shp2iECKO tumors have decreased macrophage infiltration is incorrect, since CD11b is not a specific marker of macrophages.

6. Since Sox7 is a transcription factor, it is not clear why the IF staining for SOX7 is cytoplasmic in Fig. 3i.

7. The authors conclusion that SOX7 is regulated by SHP2 has to be verified in human samples by at least showing the SOX7 levels in normal vs tumor-associated endothelial cells using human lung cancers and melanoma.

8. The authors should show the levels of SOX7 in normal and tumor-associated endothelial cells after using AAV vector to over-express SOX7.

9. Most conclusions about c-Jun and ASK1 roles in tumor angiogenesis are based on in vitro experiments or Matrigel plugs. The authors need to verify them in orthotopic mouse models of cancer and in human tissue samples.

Reviewer #3 (Remarks to the Author):

In this manuscript the authors investigated the tumor extrinsic function of SHP2 in promoting tumor growth by enhancing tumor angiogenesis and vascular abnormalization in endothelial cells. The authors used a GEMM, previously generated in their lab, in which SHP2 is selectively deleted in the adult endothelium upon tamoxifen induction. The authors found that SHP2 was a key regulator in tumor angiogenesis since its deletion was correlated with reduced tumor growth and microvascular density in mice where the tumor formation was previously induced by subcutaneous injection of two different syngeneic cancer cell lines. Similar results were also obtained in vitro using one human endothelial cell line. Mechanistically, they proposed that SHP2 promotes tumor angiogenesis not through the canonical activation of the MAPK pathway but rather the formation of a new molecular proangiogenic axis. Specifically, the authors concluded that ASK1 protein stability is enhanced by the tyrosine phosphatases function of SHP2 that in turn promoted c-Jun activation and increase the expression of the proangiogenic factor SOX7 in tumor associated endothelial cells. While SHP2 has been identified as a regulator of tumor angiogenesis (Fedele et al. Cancer Discovery 2018 and JEM 2021), this manuscript is the first to report that specific genetic deletion in adult endothelial cells promotes endothelial cell death and the involution of tumor vessels resulting in reduced tumor growth. However, more study are necessary to understand long-term effect of SHP2 deletion/inhibition in the resting vasculature of normal tissues. The current identification of SHP2 as a previously unrecognized regulator of the tumor vasculature may be a finding of exceptional relevance since the development of several SHP2 inhibitors currently in phase 1/2 trials. The manuscript is of potential interest; however, main concerns are present since some experiments are not well controlled, using few in vitro and in vivo models and sometimes the conclusions are not fully supported by the presented results.

MAJOR points:

- 1) The authors used only HUVEC cell line as in vitro model. Keys results should be validated in other endothelial cell lines as HDMEC and/or BMEM
- 2) The effect of SHP2 deletion in TME need more characterization. Only two syngeneic cell lines were used to generate tumors in the iECKO mouse model. Keys results in fig 1 and 2 should be validated using other models with a bona fide tumor microenvironment (e.g. orthotopic implantation of mammary 4T1, pancreas/lung KPC, melanoma YUMM and or D4M3A)
- 3) In fig. S1a the author showed a not complete KO of SHP2 upon tamoxifen induction in the iECKO model. Is a partial downregulation of SHP2 enough to drive the reduction in tumor growth and promote vascular normalization? Further characterization using allosteric inhibitor in vivo is necessary. Parallel evaluation of tumor sizes and vascular normalization should be performed in immunocompromised mice (e.g NSG or nude) implanted with tumor cells whose growth is SHP2-independent (e.g. ectopic expression of drug-resistant PTPN11 mutant SHP2T253M/Q257L) and

treated with different doses of the SHP2 inhibitor (e.g. SHP099 75, 37.5 and 18.75 mg/kg).

4) How the standard anti VEGF treatment (e.g. bevacizumab) in mouse models compare with SHP2 deletion/inhibition in reducing tumor growth and promoting vascular normalization?

5) In fig 3, 5 and 6, in vitro rescue experiments of the shSHP2 effect should be performed to exclude any off-target effect of the shRNA. Does the co-delivery of shSHP2 and a shRNA-resistant SHP2 cDNA rescues the SOX-7 downregulation (fig. 3), cJun and ASK reduced phosphorylation (fig. 5 and 6)? Rescue validation should be conducted in experiments in which shSOX7 and shASK1 were used as well (fig S2 and Fig 6)

6) The authors claimed that the phosphatases activity of SHP2 is important to promote ASK1 stability.

Not enough evidences are provided in fig.6 to support this conclusion. In vitro rescue experiments in which co-delivery of shSHP2 and a shRNA-resistant phosphatases-dead SHP2 cDNA (e.g. C459S mutant) should be performed. IP of ASK1 followed by pTyr WB should be performed as well using the SHP2 inhibitor in cells expressing either WT or the inhibitor-resistant mutant SHP2T253M/Q257L.

7) Based on the results provided in fig 5 a-b, the authors claimed that in endothelial cells, SHP2 promotes tumor angiogenesis with a MAPK-independent mechanism. However, not enough evidences are provided to support this conclusion. Contrasting data are also provided in fig. S3 j in which HUVEC cells treated with the SHP2i showed decreased level of pERK as well as phospho and total cJun. A better characterization is needed. Experiment using different MAPKi (e.g. SHP2, MEKi or ERKi) should be performed in vitro. Proliferation, migration assay should be performed. WB analysis of SOX7, phospho cJun, ASK1, pERK etc. should be performed as well.

Minor points:

1) Concentration and duration of experiments conducted with RMC-4550 inhibitors were not reported either in the text or in figure legends.

2) Quality of many WB should be improved (e.g. Fig 3f; Fig 5 a, b, g, h; Fig S1f; fig S3 g)

3) Fig. S3 a-g are mentioned before fig. S2 g-n

4) Graphs in fig 4d-e and j-m miss the legends

5) Missing of relevant references in the intro section. After ref # 18 Ahmed, T.A et al. Cell rep 2019, Nichols, R.J., et al. Nature Cell Bio 2018 should be included. Quintana et al. Cancer Res, 2020 and Wang, Y., et al. Sci Rep, 2021 should be discussed after ref #22 as well

Point-by-point response to the reviewers' comments

Reviewer #1 (Remarks to the Author)

The manuscript by Xu et al describes a novel and very unexpected anti cancer effect of SHP2/PTPN11 inhibitors. These drugs are considered to have anti cancer effects by blocking signals from receptor tyrosine kinases (RTKs) to downstream effectors. As such, most trials with these drugs are focussing on RTK mutant cancers. The present manuscript demonstrates that SHP2 also has a critical role in the endothelium of the cancer; as they demonstrate that deletion of SHP2 in only endothelial cells has a major effect on tumor growth. As such, these findings shed a new light on how SHP2 inhibitors can inhibit cancer growth. This finding will without doubt be of interest to a wide readership.

The biological data are supported by mechanistic studies that point towards a major role for SOX7 downstream of SHP2 in endothelial cell function.

Reply. We appreciate the positive remarking. The anti-cancer function of SHP2 inhibitors, especially combining with other anti-tumor drugs, has been well demonstrated recently. Here we show dramatic anti-cancer function of SHP2 deletion in tumor endothelial cells. Together with two recent studies revealing anti-angiogenic function of SHP2 inhibitors, our results using genetic modified mice and multiple mouse tumor models support to target SHP2 for anti-angiogenic therapy in cancers. In addition, we present a new signaling axis, ASK1-c-Jun-SOX7, as a major downstream pathway for SHP2 in regulating tumor angiogenesis. Also, our results suggest SHP2-independent tumors could also benefit from SHP2 inhibitors. SHP2 deletion improves blood perfusion in tumors which highlights the necessity in combination with other anti-tumor drugs.

1. The manuscript should be corrected by a person that is fluent in English.

Reply. Thanks for the comment. We have read through the manuscript. In addition, the writing is improved by the *Editage* (www.editage.cn).

2. Figure 1h needs a legend for the staining.

Reply. Thanks for pointing out this question. Figure 1h is H&E staining and necrosis areas are draw

out by pink staining due to decreased or vanished nuclei. In figure legends of revised version, we added the staining and measuring methods.

3. Figure 3a: why focusing only on downregulated genes? The upregulation of VEGFR is also quite strong and interesting, but it is not even discussed.

Reply. Thanks for the comments. To identify mechanisms for SHP2 in regulating tumor angiogenesis, angiogenesis-related genes were checked in SHP2-knockdown HUVECs (Figure 3a for the old version, and Figure 4a for the revised version). Multiple pro-angiogenic factors including VEGFA, VEGFR1, and VEGFR2 were upregulated upon SHP2-knockdown, which was not in line with decreased angiogenesis in SHP2 deleted mice and SHP2 knockdown endothelial cells. It is possible that the upregulation of these factors is due to the compensation to reduced angiogenesis. In the other hand, proangiogenic factor, SOX7, was significantly reduced. SOX7-knockdown in endothelial cells impaired cell proliferation, migration, and tube formation, suggesting its pro-angiogenic function. Importantly, re-expression of SOX7 in SHP2-knockdown and SHP2 deleted endothelial cells restored endothelial function and tumor angiogenesis respectively, emphasizing the critical importance of SOX7 for SHP2 in regulating tumor angiogenesis. In the Results section of the revised manuscript, we added the description that we focused on the downregulated genes because of the reduced angiogenesis.

4. Figure 4: How was the AAV vector delivered? I can guess intranasal or intratracheal injection, but it is not mentioned

Reply. Thanks for bringing out the question. The AAV expressing SOX7 was intra-tumor injected once for the viral content of 5×10^{10} TU/injection. The information was added in figure legend (Supplementary Fig. 6i) in the revised version.

5. The rationale behind looking at cJun as SOX7 regulator is unclear.

Reply. Thanks for bringing out the question. First, SOX7 is among the four transcriptional factors for SOX7 predicted in three different databases. Secondary, SHP2 is well demonstrated in regulating MAPK pathways, which including RAS-ERK, p38, and JNK-c-Jun. Subsequently, we found that in endothelial cells, SHP2 knockdown did not affect the activation of ERK and p38. On the contrary,

c-Jun was dramatically deactivated upon SHP2 knockdown or inhibition in endothelial cells. Finally, we identify SHP2-ASK1-c-Jun-SOX7, a novel pro-angiogenic signaling axis in tumor associated endothelial cells. Therefore, c-Jun mediates the regulation of SOX7 by SHP2.

Reviewer #2 (Remarks to the Author)

In this manuscript, Zhang et al. investigated the role of SHP2 in tumor-associated angiogenesis. The authors concluded that tumor-associated endothelial cells have increased levels of total and activated SHP2 that is required to de-phosphorylate and stabilize ASK1, activate c-Jun and consequently increase the SOX7 transcription. SHP2 role in endothelial cell functions has been extensively studied and SHP2 has been already implicated in regulation of vessel stability and permeability via different molecular mechanisms. Also, it was already shown that inhibition of SHP-2 suppresses angiogenesis in vitro and in vivo. The noteworthy result from the current manuscript is the new molecular mechanism implicating SHP2 in increasing the expression of SOX7 in endothelial cells. Even though the mechanism is somewhat experimentally defined, mostly in vitro, additional evidence needed and specified below. Also, it is unclear how the new mechanism is related to the previously shown role of SHP2: the authors and others have previously shown that SHP2 depletion caused endothelial cell junction disruption and lung edema. Clinical relevance of the conclusions is weak and needs to be clarified: is it relevant to the specific types of cancer or ubiquitous to all cancers.

Reply. Thanks for the comments. The function of SHP2 in endothelial cells, including angiogenesis has been well studied. Recently, it has been reported that SHP2 inhibitor inhibit tumor angiogenesis, suggesting tumor extrinsic function of SHP2 is important for tumor growth. We demonstrate endothelial SHP2 regulates tumor angiogenesis and vascular abnormalization by using Shp2 knockout and several mouse tumor models. In agree with SHP2 inhibitor, Shp2 deletion in endothelial cells inhibits tumor angiogenesis. However, tumor vascular normalization observed in tumors in Shp2 deleted mice is not induced by SHP2 inhibitors. SHP2 in non-endothelial cells in tumor microenvironment may contribute tumor vascular abnormalization.

SOX7 as a critical effector for SHP2 in regulating tumor angiogenesis and vascular abnormalization was well demonstrated by rescue experiments both in vitro and in vivo. Through ASK1-c-Jun

signaling axis, SHP2 regulates SOX7 expression in endothelial cells. All components in SHP2-ASK1-c-Jun-SOX7 signaling pathway were upregulated or activated in endothelial cells in human NSCLC tissues, emphasizing the importance of this signaling pathway in regulating tumor angiogenesis.

The importance of SHP2 in regulating tumor angiogenesis is demonstrated by two syngeneic mouse tumor models using LLC and B16 cancer cells and one orthotopic breast cancer model using E0771 cells. In addition, SHP2 inhibitor also reduces tumor angiogenesis and growth. Comparable with anti-angiogenic cediranib, SHP2 inhibitor suppressed tumor growth, especially in SHP2-independent tumors. All these results, together with other group's excellent studies, demonstrate that SHP2 inhibitors are excellent anti-angiogenic drugs in cancer therapy.

1. Since endothelial cells in different organs are phenotypically and functionally different, the authors need to use orthotopic tumor models to assess tumor angiogenesis, which is also tissue specific. They need to inoculate LLC tumor cells into the lung tissue and B16 tumor cells intradermally to make the clinically relevant models of these two types of cancer.

Reply. Thanks for the good suggestion. To increase clinical relevance of our studies, an orthotopic mouse model using E0771 mouse mammary tumor cells was introduced. As shown in Supplementary Fig. 1 and 4, SHP2 deletion in an orthotopic mouse tumor model significantly reduced tumor growth and angiogenesis and promoted vascular normalization. Therefore, SHP2 deletion displayed great anti-angiogenic function in multiple preclinical mouse tumor models.

2. In their previous publication, the authors demonstrate that conditional deletion of Shp2 in endothelial cells (EC) caused embryonic lethality associated with disruption of endothelial cell junctions and massive hemorrhage, and the inducible deletion of Shp2 in EC (Shp2iECKO) in adult mice resulted in lung vessel permeability. It is essential to measure lung vessel permeability in Shp2iECKO mice that are used for the experiments in this manuscript and their survival. Structural and functional changes in lung endothelial cells should be assessed at different time points after tamoxifen treatment, including at the time of tumor harvest, and whether those changes are similar or different in lung vs subcutaneous endothelial cells.

Reply. Constitutive deletion of Shp2 in endothelial cells led to mouse embryonic lethality. Therefore,

we construct an inducible and endothelium-specific Shp2 knockout mouse model (Shp2^{iECKO}). No overt phenotype was observed in Shp2^{iECKO} mice. Our previous studies showed that Shp2 deletion in adult mouse endothelium increased permeability in lung microvessels upon LPS treatment (Zhang J, et al. FASEB J, 2019)¹. H&E staining showed normal morphology in various organs in Shp2^{iECKO} mice including lungs, spleens, kidneys, livers, and hearts (Supplementary Fig. 3). Shp2 deletion also did not affect microvessel density in lungs, spleens, kidneys, and livers (Supplementary Fig. 3). With or without tumor burden, Shp2 deletion didn't alter vessel permeability in lungs, livers, and kidneys (Supplementary Fig. 3). These data showed that SHP2 inhibition is somehow safe in cancer therapy. It is worth mentioning that Shp2 deletion disrupted endothelial barrier in mouse embryos, while increasing endothelial barrier in tumor vessels. These results suggest different roles of SHP2 in regulating physiological and pathological angiogenesis.

3. The conclusion that SHP2 is increased and activated in tumor-associated endothelial cells in vivo is not based on strong evidence. The immunofluorescent images of mouse, and especially human lung cancer tissues (Fig. 1b-c) are of low quality and must be replaced. High and low magnifications of IF staining should be added. It should be also clarified how the relative p-SHP2 or SHP2 were quantified using these images. In addition, it is unclear why the authors use lung EC to show protein levels of SHP2 (Fig. 1b-c), but mammary EC to show mRNA levels of SHP2 (Fig. 1d).

Reply. Thanks for bringing out the questions. We replaced the representative images for SHP2 and p-SHP2 immunofluorescence staining in Fig. 1b, c with high-quality ones. Quantitative data were measured by using CD31 to label endothelial cells and the Image J software for measuring. In Fig. 1d we showed increased SHP2 mRNA in tumor associated endothelial cells. The result was extracted from the GEO dataset GSE118904, which used endothelial cells in mouse normal mammary glands and orthotopic mouse E0771 mammary tumors for single-cell RNA sequencing². The dataset GSE118904 was used here because we couldn't find similar data for lung cancer or other cancers. NSCLC tissues and conditioned media were used in our studies; however, SHP2 regulates tumor angiogenesis and vascular abnormalization shouldn't be unique to lung cancers. We didn't answer how SHP2 is upregulated and activated in tumor associated endothelial cells. This is a limitation in our studies.

4. *The statement that there are more α -SMA-positive endothelial cells in Shp2^{iECKO} vessels does not make any sense since there are no such cells.*

Reply. Thanks for bringing out the incorrect description. α -SMA is expressed in pericytes, not in endothelial cells. We revise the description. There were more endothelial cells surrounded with α -smooth muscle actin (α SMA) positive cells in tumor vessels of Shp2^{iECKO} mice, indicating increased pericyte coverage.

5. *The statement that Shp2^{iECKO} tumors have decreased macrophage infiltration is incorrect, since CD11b is not a specific marker of macrophages.*

Reply. Thanks for pointing out the question. “The infiltration of CD11b positive myeloid cells” was used to replace with the old one.

6. *Since Sox7 is a transcription factor, it is not clear why the IF staining for SOX7 is cytoplasmic in Fig. 3i.*

Reply. Thanks for bringing out the question. We did immunofluorescence staining for SOX7 in LLC tumors (Fig. 4i and supplementary Fig. 6j), NSCLC tissues (Fig. 4j), HUVECs (supplementary Fig. 5e) and Matrigel plugs (supplementary Fig. 5i). SOX7 is nuclear localization in HUVECs, in endothelial cells in LLC tumors and NSCLC tissues. SOX7 staining in the plugs was not exactly nuclear and non-specific staining is possible due to Matrigel. Same SOX7 antibody was used for all staining.

7. *The authors conclusion that SOX7 is regulated by SHP2 has to be verified in human samples by at least showing the SOX7 levels in normal vs tumor-associated endothelial cells using human lung cancers and melanoma.*

Reply. Thanks for the suggestion. As suggested, we did immunofluorescence staining for SOX7 in human NSCLC tissues and paired adjacent normal tissues (Fig. 4j). SOX7 was highly expressed in tumor vessels. Compared with normal endothelial cells, SOX7 was increased in tumor associated endothelial cells.

8. *The authors should show the levels of SOX7 in normal and tumor-associated endothelial cells*

after using AAV vector to over-express SOX7.

Reply. Thanks for the suggestion. As suggested, we did immunofluorescence staining for SOX7 in tumors in the rescue experiments (Supplementary Fig. 6j). SOX7 was decreased in tumors in Shp2iECKO mice. AAV-SOX7 injection increased SOX7 in Shp2iECKO mice. Notably, no virus-mediated SOX7 was expressed in Shp2f/f mice without Cre recombinase. The ZsGreen signal also supported the effectively expression of AAV in tumors.

9. Most conclusions about c-Jun and ASK1 roles in tumor angiogenesis are based on in vitro experiments or Matrigel plugs. The authors need to verify them in orthotopic mouse models of cancer and in human tissue samples.

Reply. Thanks for the suggestions. ASK1 expression was increased in endothelial cells in LLC tumors in Shp2iECKO mice (Fig. 7g) and in NSCLC tissues (Fig. 7i). Phospho-c-Jun was increased in endothelial cells in LLC tumors in Shp2iECKO mice (Fig. 6d). We tried to do immunofluorescence staining for phospho-c-Jun in NSCLC tissues and failed to have high-quality results. Thus, with the new data, all components in the pro-angiogenic signaling axis SHP2-ASK1-c-Jun-SOX7 are upregulated or activated in tumor associated endothelial cells.

Reviewer #3 (Remarks to the Author)

In this manuscript the authors investigated the tumor extrinsic function of SHP2 in promoting tumor growth by enhancing tumor angiogenesis and vascular abnormalization in endothelial cells. The authors used a GEMM, previously generated in their lab, in which SHP2 is selectively deleted in the adult endothelium upon tamoxifen induction. The authors found that SHP2 was a key regulator in tumor angiogenesis since its deletion was correlated with reduced tumor growth and microvascular density in mice where the tumor formation was previously induced by subcutaneous injection of two different syngeneic cancer cell lines. Similar results were also obtained in vitro using one human endothelial cell line. Mechanistically, they proposed that SHP2 promotes tumor angiogenesis not through the canonical activation of the MAPK pathway but rather the formation of a new molecular proangiogenic axis. Specifically, the authors concluded that ASK1 protein stability is enhanced by the tyrosine phosphatases function of SHP2 that in turn promoted c-Jun

activation and increase the expression of the proangiogenic factor SOX7 in tumor associated endothelial cells. While SHP2 has been identified as a regulator of tumor angiogenesis (Fedele et al. Cancer Discovery 2018 and JEM 2021), this manuscript is the first to report that specific genetic deletion in adult endothelial cells promotes endothelial cell death and the involution of tumor vessels resulting in reduced tumor growth. However, more study are necessary to understand long-term effect of SHP2 deletion/inhibition in the resting vasculature of normal tissues. The current identification of SHP2 as a previously unrecognized regulator of the tumor vasculature may be a finding of exceptional relevance since the development of several SHP2 inhibitors currently in phase 1/2 trials. The manuscript is of potential interest; however, main concerns are present since some experiments are not well controlled, using few in vitro and in vivo models and sometimes the conclusions are not fully supported by the presented results.

Reply. Thanks for the positive remarking. Since the findings of allosteric SHP2 inhibitors, the anti-tumor role of SHP2 inhibitors has been extremely studied. Recently, the tumor extrinsic function of SHP2, especially the anti-angiogenic function has been suggested and well demonstrated (Fedele et al. Cancer Discovery 2018 and JEM 2021; Wang Y et al. EMBO Mol Med 2021; Tang KH, et al. Cancer Discov 2021)^{3, 4, 5, 6}. Using an endothelium-specific knockout mouse model, we clearly demonstrate that SHP2 in tumor associated endothelial cells contributes greatly in tumor angiogenesis and growth. This result further supports SHP2 inhibitors for anti-tumor therapy, even SHP2-independent tumors.

To further consolidate our conclusions, as suggested by the reviewers, orthotopic mouse mammary cancer model was introduced. Similar to the other two mouse tumor models, Shp2 deletion greatly impaired tumor angiogenesis, growth and tumor vascular abnormalization. Most tumors are dependent on vessels for growth and metastasis; therefore, SHP2 is a potential target in anti-angiogenic therapy in cancers. Tumor vascular normalization by Shp2 deletion enhances the rationale for combination treatments.

At last, by the successful rescue experiments, SOX7 is the critical downstream effector for SHP2 in regulation tumor angiogenesis. SHP2 interacts and stabilizes ASK1, making SHP2-ASK1-c-Jun-SOX7 as an important pro-angiogenic signaling axis in tumor associated endothelial cells. Altogether, using gene deletion mouse model and pharmacological inhibitor, we show significant anti-angiogenic and anti-tumor roles of SHP2 deletion and inhibition in multiple mouse tumor

models. Moreover, we present a novel pro-angiogenic signaling axis SHP2-ASK1-c-Jun-SOX7, which is upregulated and activated in tumor associated endothelial cells in human cancers.

MAJOR points:

1) The authors used only HUVEC cell line as in vitro model. Keys results should be validated in other endothelial cell lines as HDMEC and/or BMEM.

Reply. Thanks for the suggestion. As suggested, we used hCMEC (human cerebral microvessel endothelial cell) as another endothelial cells line (Supplementary Fig. 2). Similar to those in HUVECs, SHP2 knockdown in hCMEC impaired cell proliferation, migration and tube formation. Together with the results we gained in primary lung endothelial cells isolated from Shp2^{iECKO} mice, SHP2 is essential for endothelial function.

2) The effect of SHP2 deletion in TME need more characterization. Only two syngeneic cell lines were used to generate tumors in the iECKO mouse model. Keys results in fig 1 and 2 should be validated using other models with a bona fide tumor microenvironment (e.g. orthotopic implantation of mammary 4T1, pancreas/lung KPC, melanoma YUMM and or D4M3A).

Reply. Thanks for the suggestions. As suggested, we introduced orthotopic mouse mammary tumor model using E0771 mammary cancer cells (Supplementary Fig. 1, 4). Similar to the results in other two mouse tumor models, SHP2 deletion reduced tumor angiogenesis and growth, as well as tumor vascular abnormalization. Thus, the anti-angiogenic function of SHP2 inhibition is not specific to certain tumors, while being specific to basic regulating mechanisms in tumor vessels.

3) In fig. S1a the author showed a not complete KO of SHP2 upon tamoxifen induction in the iECKO model. Is a partial downregulation of SHP2 enough to drive the reduction in tumor growth and promote vascular normalization? Further characterization using allosteric inhibitor in vivo is necessary. Parallel evaluation of tumor sizes and vascular normalization should be performed in immunocompromised mice (e.g NSG or nude) implanted with tumor cells whose growth is SHP2-independent (e.g. ectopic expression of drug-resistant PTPN11 mutant SHP2T253M/Q257L) and treated with different doses of the SHP2 inhibitor (e.g. SHP099 75, 37.5 and 18.75 mg/kg).

Reply. Thanks for suggestions. Constitutively Shp2 knockout in mouse endothelium leads to

embryonic lethality; therefore, we constructed this inducible and endothelial-specific Shp2 knockout mouse model (Shp2iECKO). In inducible and conditional gene knockout mouse, gene is completely disrupted in most but not all endothelial cells. This common phenome is caused by the efficiency of tamoxifen induction, Cre recombinase shuttle from cytosol to nucleus, and other factors.

As suggested, we constructed inhibitor-resistant SHP2 mutant (SHP2T253M/Q257L). in SHP2-knockdown LM3 cells, SHP2 wildtype and mutant were re-expressed respectively. As previously reported, SHP2 inhibitor SHP099 significantly inhibited tumor growth. Notably, tumor growth was also inhibited in LM3 cells expressed inhibitor-resistant SHP2 mutant. The inhibition efficiency was comparable in two groups, suggesting that the tumor promoting function of SHP2 is mostly tumor extrinsic in this mouse tumor model. In addition, SHP2 inhibitor showed comparable anti-tumor effect to VEGFR inhibitor cediranib. All these data suggest that anti-angiogenic function contributes most of the anti-tumor effect of SHP2 inhibitor.

Both Shp2 deletion and inhibition reduced angiogenesis and tumor growth. Pericyte coverage, increased in Shp2 deletion, was reduced in SHP2 inhibitor (Fig. 3g and Fedele et al. J Exp Med 2021; Wang Y et al. EMBO Mol Med 2021)^{4,5}. The interesting difference suggests that SHP2 plays functional role in pericytes. Our previous studies showed Shp2 deletion in vascular smooth muscle cells results in defects in vascular development and embryonic lethality (Gong H, et al. J Mol Cell Cardiol 2019)⁷. SHP2 is involved in the PDGF-BB/PDGFR signaling, which is required for vascular maturation.

4) How the standard anti VEGF treatment (e.g. bevacizumab) in mouse models compare with SHP2 deletion/inhibition in reducing tumor growth and promoting vascular normalization?

Reply. Thanks for the suggestions. In the figure 3, cediranib, a VEGFR inhibitor was used in a mouse tumor model. Our results showed that cediranib (1.5 mg/kg) exhibited anti-tumor effect. SHP099 was comparable to cediranib in reducing tumor growth, which supports the anti-angiogenic function of SHP2 inhibition.

VEGFA-VEGFR signaling pathway is the major pro-angiogenic signal in endothelial cells. In clinician, anti-angiogenic therapy is only effective in a few tumors. Drug resistance is a major limitation. SHP2 mediates multiple receptor tyrosine kinase (RTK) signals in both cancer cells and

endothelial cells. Targeting SHP2 will be more effective in anti-angiogenesis, compared with separately targeting VEGFA-VEGFR, or FGF-FGFR. Moreover, synergistic anti-tumor effect of SHP2 inhibitors and inhibitors for MAPK pathway such as MEK inhibitors has been well demonstrated in cancer cells in preclinical tumor models. It is rationale that SHP2 inhibitor, combining with existing drugs targeting VEGFR pathway will exhibit excellent anti-tumor effect and eliminate drug resistance. These are worth further investigations.

5) In fig 3, 5 and 6, in vitro rescue experiments of the shSHP2 effect should be performed to exclude any off-target effect of the shRNA. Does the co-delivery of shSHP2 and a shRNA-resistant SHP2 cDNA rescues the SOX-7 downregulation (fig. 3), cJun and ASK reduced phosphorylation (fig. 5 and 6)? Rescue validation should be conducted in experiments in which shSOX7 and shASK1 were used as well (fig S2 and Fig 6).

Reply. Thanks very much for the suggestions. As suggested, in both HUVECs and hCMECs, SHP2 was re-expressed in SHP2-knockdown cells (Supplementary Fig. 2). All defects caused by SHP2 knockdown including cell proliferation, migration, and tube formation in endothelial cells were restored by SHP2 re-expression. To knockdown SHP2, targeting sequence was designed in the untranslated region (utr) in SHP2 mRNA. SOX7 and phosphorylated c-Jun was restored by the re-expression in SHP2 knockdown endothelial cells (Supplementary Fig. 5a, 7ba). Similarly, SOX7 was re-expressed and defects caused by SOX7 knockdown were restored (Supplementary Fig. 6a-d). ASK1 was re-expressed to restore SOX7 expression in ASK1-knockdown endothelial cells (Supplementary Fig. 8e). These new data reduced the possibility of off-target effect of shRNA and therefore enhanced our conclusion.

6) The authors claimed that the phosphatases activity of SHP2 is important to promote ASK1 stability.

Not enough evidences are provided in fig.6 to support this conclusion. In vitro rescue experiments in which co-delivery of shSHP2 and a shRNA-resistant phosphatases-dead SHP2 cDNA (e.g. C459S mutant) should be performed. IP of ASK1 followed by pTyr WB should be performed as well using the SHP2 inhibitor in cells expressing either WT or the inhibitor-resistant mutant SHP2T253M/Q257L.

Reply. Studies about SHP2 dephosphorylates and stabilizes ASK1 in endothelial cells has been well performed. Phosphorylation on Tyr718 in ASK1 is required for the interaction with E3 ligase SOCS1 and thereafter degradation (He Y, et al. JBC 2006)⁸. SHP2 dephosphorylates ASK1 on Tyr718 (Yu L, et al. JBC 2009)⁹. No commercial antibody for phosphor-ASK1 (Tyr718) is available. Thus we study the effect of SHP2 on ASK1 stability. SHP2 interacts with ASK1 (Fig. 7e). SHP2 wild-type, but not SHP2 catalytic dead mutant (C459S) restored ASK1 in SHP2-knockdown endothelial cells (Fig. 7b), suggesting phosphatase activity of SHP2 is required for ASK1 stability. In addition, SHP2 inhibitor increased ASK1 in cells expressing wildtype SHP2, but not in cells expressing inhibitor-resistant SHP2 (SHP2T253M/Q257L). The results again support the conclusion that phosphatase activity of SHP2 is essential to stabilize ASK1.

7) Based on the results provided in fig 5 a-b, the authors claimed that in endothelial cells, SHP2 promotes tumor angiogenesis with a MAPK-independent mechanism. However, not enough evidences are provided to support this conclusion. Contrasting data are also provided in fig. S3 j in which HUVEC cells treated with the SHP2i showed decreased level of pERK as well as phospho and total cJun. A better characterization is needed. Experiment using different MAPKi (e.g. SHP2, MEKi or ERKi) should be performed in vitro. Proliferation, migration assay should be performed. WB analysis of SOX7, phospho cJun, ASK1, pERK etc. should be performed as well.

Reply. Thanks a lot for bringing out these questions. It is cell-type dependent for SHP2 in regulating Ras-Erk signaling pathway. Our previous studies showed that phosphor-Erk was decreased upon SHP2 knockdown in lung cancer cells (A549) (Fig 1H, Li S, et al. J Biol Chem 2014)¹⁰. SHP2 deletion in mouse vascular smooth muscle cells did not affect phosphor-Erk, as well as phosphor-p-38, phosphor-JNK (Fig S6C-D, Gong H, et al. J Mol Cell Cardiol 2019)⁷. Phosphor-Erk was increased in SHP2 deleted mouse dendritic cells (Fig 1f, Xu Y, et al. Cell Mol Immunol 2021)¹¹. In current manuscript, phosphor-Erk didn't change upon SHP2 inefficiency in both SHP2-knockdown HUVECs and Shp2-knockout mouse lung endothelial cells (Fig. 6a-b). Therefore, the role of SHP2 in regulating Ras-Erk signaling pathway relies on cell types.

The change in phosphor-Erk in HUVECs treated with SHP2 inhibitor didn't agree with that in SHP2-knockdown HUVECs. Phospho-Erk was decreased in HUVECs treated with RMC4550 (Supplementary Fig 6c), but unchanged in SHP2-knockdown HUVECs (Fig 6a). Our results about

RMC4550 were consistent with recent published data (Wang Y, et al. EMBO Mol Med 2021)⁵. We noticed the difference and therefore these experiments were repeated many times. Same issue was happened in our ongoing project. The off-target effect of SHP2 knockdown was excluded by SHP2 re-expression. We think it is possible that SHP2 allosteric inhibitors (SHP099 and RMC4550) have other targets than SHP2. Unfortunately, we don't have evidence for this yet.

SOX7 and phosphor-c-Jun were decreased in both SHP2 inhibition and knockdown HUVECs. Our data showed MEK inhibitor AZD6244 decreased SOX7 expression without affecting c-Jun signaling in endothelial cells (Figure 1). Moreover, SHP2 knockdown in HUVECs didn't change Erk activation induced by various growth factors (Figure 2). Therefore, in endothelial cells, SOX7 expression is controlled by SHP2 via c-Jun signaling pathway.

Figure 1. Western blot for SOX7, c-Jun, ERK, and p-38 and associated phosphorylated forms in HUVECs treated with MEK inhibitor (AZD6244, 24 h). β -Actin was used as a loading control. Quantitative data were expressed as mean \pm SEM for three independent experiments. **, $p < 0.01$, ***, $p < 0.001$, by one-way ANOVA with multi-comparisons.

Figure 2. Western blot for p-Erk in SHP2 knock-down HUVECs treated with VEGF-A, bFGF, HGF, EGF (10 ng/ml) for 10 min. β -Actin was used as a loading control.

Minor points:

1) Concentration and duration of experiments conducted with RMC-4550 inhibitors were not reported either in the text or in figure legends.

Reply. Thanks for pointing out this incorrect. Various concentrations of RMC4550 were used to study endothelial function (supplementary Fig. 2a-c), SOX7 expression (supplementary Fig. 5d) and ASK1-c-Jun signaling (supplementary Fig. 7c, 8b). In a concentration dependent manner, RMC4550 decreased endothelial function and ASK1-c-Jun-SOX7 signaling.

2) *Quality of many WB should be improved (e.g. Fig 3f; Fig 5 a, b, g, h; Fig S1f; fig S3 g).*

Reply. Thanks for the suggestions. We have redone the experiments and new results were shown in the new version (*Fig 3f to Fig 4f; Fig 5a to Fig 6a; Fig 5b to Fig 6b; Fig 5g to Fig 6h; Fig 5h to Fig 6c; Fig S1f to Fig S1k; fig S3 g to Fig S5h*).

3) *Fig. S3 a-g are mentioned before fig. S2 g-n.*

Reply. Thanks for pointing out this mistake. We make a revision in the new version.

4) *Graphs in fig 4d-e and j-m miss the legends.*

Reply. Sorry for the careless errors. These errors are corrected in the new manuscript.

5) *Missing of relevant references in the intro section. After ref # 18 Ahmed, T.A et al. Cell rep 2019, Nichols, R.J., et al. Nature Cell Bio 2018 should be included. Quintana et al. Cancer Res, 2020 and Wang, Y., et al. Sci Rep, 2021 should be discussed after ref #22 as well.*

Reply. Thanks for these nice suggestions. All references are added as suggested in our new manuscript.

REFERENCES

1. Zhang J, *et al.* SHP2 protects endothelial cell barrier through suppressing VE-cadherin internalization regulated by MET-ARF1. *FASEB J* **33**, 1124-1137 (2019).
2. McCann JV, *et al.* Endothelial miR-30c suppresses tumor growth via inhibition of TGF-beta-induced Serpine1. *J Clin Invest* **129**, 1654-1670 (2019).
3. Fedele C, *et al.* SHP2 Inhibition Prevents Adaptive Resistance to MEK Inhibitors in Multiple

Cancer Models. *Cancer Discov* **8**, 1237-1249 (2018).

4. Fedele C, *et al.* SHP2 inhibition diminishes KRASG12C cycling and promotes tumor microenvironment remodeling. *J Exp Med* **218**, (2021).
5. Wang Y, *et al.* Targeting the SHP2 phosphatase promotes vascular damage and inhibition of tumor growth. *EMBO Mol Med* **13**, e14089 (2021).
6. Tang KH, *et al.* Combined Inhibition of SHP2 and CXCR1/2 Promotes Anti-Tumor T Cell Response in NSCLC. *Cancer Discov*, (2021).
7. Gong H, *et al.* Shp2 in myocytes is essential for cardiovascular and neointima development. *J Mol Cell Cardiol* **137**, 71-81 (2019).
8. He Y, Zhang W, Zhang R, Zhang H, Min W. SOCS1 inhibits tumor necrosis factor-induced activation of ASK1-JNK inflammatory signaling by mediating ASK1 degradation. *J Biol Chem* **281**, 5559-5566 (2006).
9. Yu L, *et al.* JAK2 and SHP2 reciprocally regulate tyrosine phosphorylation and stability of proapoptotic protein ASK1. *J Biol Chem* **284**, 13481-13488 (2009).
10. Li S, *et al.* SHP2 positively regulates TGFbeta1-induced epithelial-mesenchymal transition modulated by its novel interacting protein Hook1. *J Biol Chem* **289**, 34152-34160 (2014).
11. Xu Y, *et al.* Tyrosine phosphatase Shp2 regulates p115RhoGEF/Rho-dependent dendritic cell migration. *Cell Mol Immunol* **18**, 755-757 (2021).

REVIEWERS' COMMENTS

Reviewer #1 (Remarks to the Author):

The manuscript has been improved further by the recent revisions. It is suitable for publication in its present form. Use of English language can be improved. It would be excellent if the authors would ask a native speaker to go through the text before publication.

Reviewer #2 (Remarks to the Author):

My comments were addressed. The quality of the manuscript is improved.

Reviewer #3 (Remarks to the Author):

The authors comprehensively addressed all the major points of the revision. In my opinion, this work can be accepted for publication in Nature Communication journal.

Minor points:

- a) the results obtained with the E0771 orthotopic model reported in the new S1 figure should, at least in part, be moved in the main figure 1.
- b) rescue experiments in fig S2 should, at least in part, be moved in the main figure 2.
- c) in figure 3 the more generic "SHP2mut" should be replaced with the more specific "SHP2_TM/QL"

Point-by-point response to the reviewers' comments

Reviewer #1 (Remarks to the Author)

Reviewer #1 (Remarks to the Author):

1. The manuscript has been improved further by the recent revisions. It is suitable for publication in its present form. Use of English language can be improved. It would be excellent if the authors would ask a native speaker to go through the text before publication.

Reply: Thanks for the comment. We polished the manuscript once more with *editage* English language editing services. We hope this improved version can meet your requirement.

Reviewer #2 (Remarks to the Author):

1. My comments were addressed. The quality of the manuscript is improved.

Reply: We are delighted that the reviewer finds our revised manuscript improved.

Reviewer #3 (Remarks to the Author):

The authors comprehensively addressed all the major points of the revision. In my opinion, this work can be accepted for publication in Nature Communication journal.

Reply: We greatly appreciate the positive comment.

Minor points:

a) the results obtained with the E0771 orthotopic model reported in the new S1 figure should, at least in part, be moved in the main figure 1.

Reply: Thank you for the suggestion. We have moved featured results about E0771 orthotopic model, including tumor images, tumor growth and tumor weight (Supplementary Fig. 1 e-g), to the main Figure 1 (Figure 1h-j).

b) rescue experiments in fig S2 should, at least in part, be moved in the main figure 2.

Reply: Thanks a lot for your suggestion. We have moved part of the results of rescue experiments (Supplementary Fig. 2 d, e) to the main figure 2 (Figure 2f, g). To avoid the excessive length of the main figure 2, we moved data for MLEC cell proliferation and migration experiments (Figure 2e, f) to Supplementary Figure 2 (Supplementary Figure 2a, b).

c) in figure 3 the more generic "SHP2mut" should be replaced with the more specific "SHP2_TM/QL"

Reply: Thank you for the suggestion. We have replaced "SHP2mut" with "SHP2_TM/QL" as suggested.